# Geo-Mamba: Dual-Path Riemannian State-Space Models for Functional Dynamics

## Abstract

Functional magnetic resonance imaging (fMRI)–derived functional connectivity (FC) is represented as graphs and as correlation/covariance matrices that live on non-Euclidean spaces—cortical graphs and the Riemannian manifold of symmetric positive-definite (SPD) matrices—so conventional Euclidean sequence models are misspecified. To this end, we introduce *Geo-Mamba*, a geometric variant of Mamba formulated on Riemannian manifolds. *Geo-Mamba* employs a dual-path selective state-space design: *a stacked path* performs hierarchical spatial modeling by aggregating pyramid multi-scale features to capture local and global dependencies, while *a embedding path* combats redundancy in high-dimensional SPD inputs via progressive, geometry-aware dimensionality reduction (operating in the manifold spaces) to produce compact states without violating Riemannian constraints. Their complementary outputs are fused through the tailored *GeoMix* operator to yield a compact, discriminative SPD representation. *Geo-Mamba* is evaluated on six public fMRI datasets—ADNI, OASIS, PPMI, Taowu, Neurocon, and Mātai—spanning Alzheimer's and Parkinson's cohorts as well as multi-site normative populations with diverse acquisition protocols. Across these benchmarks, it delivers consistently competitive accuracy and robustness, supporting the value of dual-path manifold modeling for neuroimaging and its potential for clinical translation.

## 1 Introduction

Functional magnetic resonance imaging (fMRI) indirectly reflects neural activity by recording blood oxygen level-dependent (BOLD) signals in the brain (Bandettini et al., 1992), providing important information for revealing brain functional networks and their dynamic characteristics. Functional connectivity (FC) is usually quantified by calculating the correlation of BOLD time series among different brain regions of interest (ROIs) (Van Den Heuvel & Pol, 2010; Friston, 2011), the symmetric positive definite (SPD) matrix thus obtained is not only a "snapshot" depicting information integration but also the fundamental input for subsequent network neuroscience analysis. The overall structure of the FC matrix not only reflects pairwise correlation but also the covariance structure and dynamic stability at the network level. Therefore, it is more naturally regarded as a family of SPD matrices, and distortions can arise when Euclidean operations (e.g., naive batch averaging, interpolation and PCA) are applied without respecting the Riemannian geometry, breaking positive definiteness and disrupting the overall pattern of functional interactions You & Park (2021); Pennec et al. (2006). Based on this, FC is more suitable for modeling on SPD manifolds, where Riemannian geometry provides a consistent metric framework for characterizing its dynamic evolution. Among the numerous SPD metrics, log-Euclidean metric Arsigny et al. (2007), which linearizes the manifold through the logarithm of the matrix and combines geometric fidelity with computational controllability. Compared with affine invariant Riemann metric (AIRM) or Stein divergence, it is more suitable for the training of stable and differentiable deep models.

The state-space model (SSM) has long been used in neuroimaging analysis to depict the dynamic processes of brain activity, especially demonstrating unique advantages in fMRI research. Early studies such as Janoos et al. (2011) proposed a data-driven SSM method through Hidden Markov Model (HMM) and Monte Carlo expectation maximization algorithm Unsupervised extraction of spatiotemporal dynamic patterns related to cognitive tasks from fMRI data effectively identified potential psychological states and avoided reliance on experimental labels. Then, Suk et al. (2016)

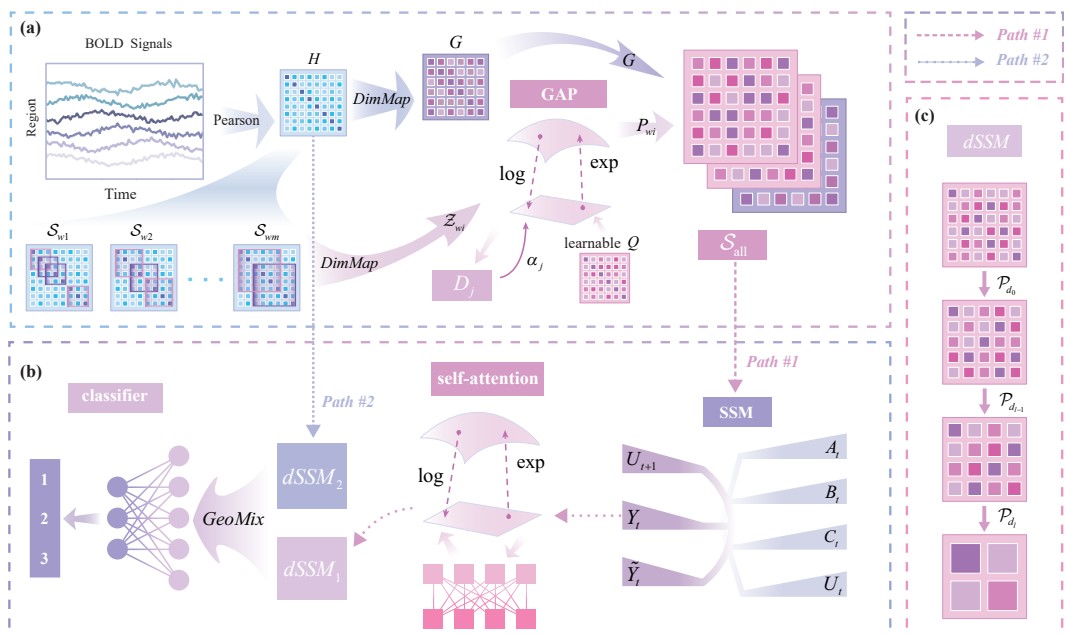

Figure 1: Overall architecture of *Geo-Mamba*. (a) Pyramid-based geometric feature representation for characterizing multi-scale brain network patterns. (b) Two key processing paths for spatial feature encoding and fusion. (c) Example of manifold-aware dimension reduction using *dSSM*, preserving high-dimensional structural information while compressing essential features.

combines the deep autoencoder with SSM. It is used for the estimation of functional dynamics in resting-state fMRI (rs-fMRI). After capturing nonlinear functional relationships in low-dimensional space, HMM is utilized for dynamic modeling, achieving the early diagnosis of mild cognitive impairment (MCI). In addition, researchers have proposed a variety of SSM variants. For example, He et al. (2023) utilizes the switching state-space model combined with variable Bayesian inference to detect transient neural dynamics. Fernandes et al. (2020) introduce the applicability and limitations of state-space Granger causality in fMRI causal inference were explored. These works collectively demonstrate that SSM can effectively model functional connectivity and cognitive processes in fMRI data, providing a powerful tool for understanding brain functional organization.

In the field of sequence modeling, the recently proposed Mamba Gu & Dao (2023) has made an important expansion of SSM. This method, by introducing the selective state space model (selective SSM), allows the model parameters to change dynamically with the input, thereby selectively remembering or forgetting information in the sequence dimension, and overcomes the defect that traditional linear or convolutional SSMs cannot perform content-dependent reasoning. Based on the efficient modeling ability of Mamba, in the last two years, many studies have introduced it into brain image analysis to explore the representation learning of complex neural time series. For example, Wei et al. (2025) proposes the *FST-Mamba* framework, which models the spatial and temporal dimensions respectively with a hierarchical structure. Combined with variable-scale aggregation and symmetrical position coding mechanisms, the analysis ability of dynamic functional network connections in fMRI has been significantly enhanced. Behrouz & Hashemi (2024) proposed two Mamba-based modules—*BTMamba* for temporal channel fusion and *BNMamba* for spatial inter-regional dependency modeling—integrated via a gating mechanism. The resulting architecture reports state-of-the-art performance on multimodal neuroimaging tasks. Li et al. (2025) is aimed at multisite rs-fMRI The multi-scale context modeling and cross-site feature alignment mechanism were proposed, effectively solving the problem of poor generalization of cross-center data, and performed outstandingly in the detection tasks of depression and autism. It should be pointed out that although these methods have achieved efficient spatial and temporal modeling by using the Mamba architecture, their modeling process is still based on Euclidean space and does not explicitly consider the Riemannian manifold structure in which the fMRI FC matrix is located.

To address the limitations of Euclidean SSMs for functional connectivity analysis, we propose *Geo-Mamba*, a geometric extension of selective SSMs on SPD Riemannian manifolds. It integrates pyramid-based multi-scale feature extraction with stacked manifold SSMs (i.e., *stacked path*), cap-

turing both local and global geometric structures while enabling hierarchical modeling. Parallel manifold-aware dimensionality reduction compresses high-dimensional SPD inputs (i.e., *embedding path*), and the outputs are fused via geodesic interpolation, preserving positive definiteness and intrinsic curvature. This design provides an efficient, interpretable framework for long-range modeling of FC dynamics of the whole brain.

## 2 METHOD

Given a time series of BOLD signals $[E_i^t]_{i=1...N}^{t=1...T} \in \mathbb{R}^{N \times T}$, where $N$ denotes ROIs and $T$ denotes the number of time points. By calculating the Pearson correlation coefficient of signals in any two brain regions in the time dimension, we can compute a FC matrix $X \in \mathbb{R}^{N \times N}$, which is used to describe the functional correlation between brain regions.

We propose a novel geometric state space model, *Geo-Mamba* (as shown in Fig. 1), aiming to characterize the functional dynamics on SPD manifolds within the Riemannian geometry framework. In Sec. 2.2, we propose a pyramid-based geometric feature extraction mechanism, which effectively captures local and global features at different levels by introducing a multi-scale sliding window in the main diagonal direction of the FC matrix. In Sec. 2.3, we extend the selective SSM framework from traditional Euclidean space to the space of symmetric positive definite (SPD) matrices, enabling state updates and dynamic modeling under the constraints of nonlinear Riemannian manifolds. Following this, Sec. 2.4 demonstrates that stacking multiple manifold-based SSM layers enables the model to effectively capture the hierarchical and multi-scale spatial evolution of brain functional dynamics.

### 2.1 PRELIMINARIES ON RIEMANNIAN GEOMETRY OF SPD MATRICES

A SPD matrix is defined as an $n \times n$ real matrix $X$ that satisfies the condition $\beta^T X \beta > 0$ for all non-zero real vectors $\beta$. In fMRI analysis, the Pearson correlation coefficient matrix can be regarded as a SPD matrix that encodes the geometric patterns of functional connectivity. The collection of all $n \times n$ SPD matrices constitutes a smooth Riemannian manifold $\mathcal{M}$. Due to the manifold's non-linearity, Riemannian operations are required to preserve its geometry and avoid distortions from Euclidean methods. According to the logarithmic Euclidean framework (Arsigny et al., 2007), the tangent space $\mathcal{T}_Y \mathcal{M}$ of the manifold $\mathcal{M}$ at a point $Y \in \mathcal{M}$ is defined as the set of all vectors tangent to the manifold at $Y$. Since the tangent space possesses Euclidean properties, standard linear algebraic operations (like addition and linear transformation) can be readily performed in $\mathcal{T}_Y \mathcal{M}$, which provides computational convenience after logarithmic mapping. To establish a correspondence between the manifold and its tangent space, the logarithmic and exponential mappings play a central role. Specifically, the logarithmic mapping $\log : \mathcal{M} \to \mathbb{R}_{\text{sym}}^{n \times n}$ projects a SPD matrix onto the tangent space, thereby enabling Euclidean processing, while the exponential mapping $\exp : \mathbb{R}_{\text{sym}}^{n \times n} \to \mathcal{M}$ reverses this projection, mapping tangent vectors back onto the manifold while preserving its intrinsic geometric structure.

For any SPD matrix $X \in \mathcal{M}$, the eigenvalue decomposition is given by $X = V \Lambda V^\top$, $\Lambda = \text{diag}(\lambda_1, \ldots, \lambda_n)$, $\lambda_i > 0$, where $V$ is an orthogonal matrix of eigenvectors and $\Lambda$ is a diagonal matrix of strictly positive eigenvalues. Based on this decomposition, the logarithmic and exponential mappings on the SPD manifold are defined as

$$\log(X) = V \log(\Lambda) V^\top, \quad \exp(X) = V \exp(\Lambda) V^\top, \tag{1}$$

where $\log(\Lambda)$ and $\exp(\Lambda)$ denote element-wise operations on the diagonal. The logarithmic map projects $X$ onto the tangent space by applying the logarithm to its eigenvalues, while the exponential map lifts a symmetric matrix back onto the SPD manifold.

### 2.2 PYRAMID-BASED GEOMETRIC FEATURE REPRESENTATION

To capture both multi-scale local geometric structures and global topological information on the SPD manifold, we propose a pyramid-based geometric feature extraction module, as shown in Fig. 1 (a). This module performs multi-scale sub-block partitioning and mapping on the input SPD matrix sequence, generating feature representations with rich multi-scale geometric semantics. These

representations are further integrated with global features to construct a unified feature sequence, which serves as a *multi-scale feature sequence* for subsequent modeling.

Since FC matrices are typically ordered by brain regions, the diagonal regions of the matrix inherently capture localized connectivity patterns within spatially or functionally adjacent brain areas ((Pinotsis et al., 2013; Shen et al., 2015)). Extracting diagonal sub-blocks thus corresponds to isolating interactions among contiguous brain regions, which reflect localized network modules. By applying sliding windows of different sizes, we are able to capture multi-scale local dependencies ranging from fine-grained regional clusters to broader cortical sub-networks, while preserving the SPD structure of the sub-blocks. Given an input SPD matrix $H \in \mathbb{R}^{n \times n}$, for a predefined set of window sizes $w_1, w_2, \ldots, w_m$, we extract all possible $w \times w$ diagonal sub-blocks as

$$\mathcal{S}_w = \{H_{:,i:i+w,i:i+w} \mid i = 1, 2, \ldots, n - w + 1\}, \quad \mathcal{S}_w \in \mathbb{R}^{K_w \times w \times w}, \tag{2}$$

where $K_w = n - w + 1$ denotes the number of sub-blocks obtained by the sliding window of size $w$. For each sub-block $S \in \mathcal{S}_w$, we apply a *DimMap* projection that geometrically compresses its representation to the state dimension $d \ll w$. $Z = W^\top S W$, $W \in \mathbb{R}^{w \times d}$, where $W \in \mathbb{R}^{w \times d}$ is an orthogonal parameter matrix defined on the Stiefel manifold (Dan et al., 2022b), ensuring that the mapped matrix $Z$ remains SPD. The collection of mapped sub-blocks is denoted as $\mathcal{Z}_w = \{Z_j\}_{j=1}^{K_w}$, where $\mathcal{Z}_w \in \mathbb{R}^{K_w \times d \times d}$.

To aggregate the sub-block features $Z_w$ at scale $w$, we propose a *geometric attention pooling* (GAP) module that operates in the Log-Euclidean space, ensuring compatibility with the Riemannian geometry of SPD matrices. The module proceeds as follows:

$$D_j = \log(Z_j) - Q, \quad \alpha_j = \frac{\exp\left(-\|D_j\|_F^2\right)}{\sum_{k=1}^{K_w} \exp\left(-\|D_k\|_F^2\right)}, \quad P_w = \exp\left(\sum_{j=1}^{K_w} \alpha_j \cdot \log(Z_j)\right) \tag{3}$$

where $j = 1, \ldots, K_w$, $Q \in \mathbb{R}^{d \times d}$ is a learnable symmetric matrix that acts as a geometric anchor in the tangent space, each window scale maintains its own GAP module and an independent $Q$, enabling scale-specific geometric modeling. The logarithmic and exponential maps (defined in Eq. 1) transform SPD matrices between the manifold and its tangent space via eigenvalue decomposition.

By projecting each $Z_j$ into the tangent space and measuring its distance from $Q$, the GAP module computes attention weights $\alpha_j$ that reflect the geometric relevance of each sub-block. The final aggregated representation $P_w$ is obtained through a weighted Fréchet-type mean in the log domain, followed by an exponential map that returns the result to the SPD manifold. This design enables geometry-aware attention and pooling, while preserving the intrinsic structure of the SPD space.

After obtaining the aggregated representations $\{P_{w_1}, \ldots, P_{w_m}\}$ from multiple scales in the local pyramid, we introduce a *global contextual feature* by performing matrix-based dimensionality reduction on the global feature map $H$: $G = W_g^\top H W_g$, $G \in \mathbb{R}^{d \times d}$, where $W_g \in \mathbb{R}^{n \times d}$ is a learnable projection matrix. The local features $\{P_{w_1}, \ldots, P_{w_m}\}$ and the global descriptor $G$ are then concatenated along a new axis to form a unified representation:

$$\mathcal{S}_{\text{all}} = [P_{w_1}, P_{w_2}, \ldots, P_{w_m}, G], \quad \mathcal{S}_{\text{all}} \in \mathbb{R}^{(m+1) \times d \times d}. \tag{4}$$

This consolidated structure is treated as a *multi-scale feature sequence* and subsequently fed into the manifold-based SSM module. This design enables the modeling of dynamic evolution patterns and significantly enhances the spatial representation capacity of non-Euclidean geometric features.

## 2.3 GEOMETRIC SELECTIVE STATE SPACE MODELS ON RIEMANNIAN MANIFOLDS

In this section, we propose the Riemannian selective state space model (*R-SSM*), a geometric extension of the traditional selective SSM to Riemannian manifolds for modeling sequences of SPD matrices. *R-SSM* inherits the core structure of conventional SSMs while adapting its computations to the non-Euclidean geometry of the SPD manifold. To enhance flexibility, *R-SSM* optionally incorporates a *DimMap* dimensionality reduction mechanism: (i) When no reduction is applied, the model reduces to the standard SSM, which performs state updates directly on the full-dimensional SPD inputs. (ii) When dimensionality reduction is enabled, the resulting variant—denoted as *dSSM*—first projects the input SPD matrices into a lower-dimensional SPD subspace using *DimMap* before proceeding with the Riemannian SSM updates. In both settings, all state transitions and outputs are

computed on the SPD manifold, ensuring full geometric consistency. The design of *R-SSM* follows three core constraints:

**Manifold constraints and parameter generation.** At any spatial window step, both the input *multi-scale feature sequence* $\mathcal{S}_{\text{all}}$ and the hidden state $U_t$ are strictly confined within the SPD manifold $\mathcal{M}$ to maintain geometric consistency and numerical stability. The hidden state is initialized close to zero: $U_0 = \epsilon I_d$, $\epsilon = 1e^{-3}$, ensuring that the initial state is SPD and suitable for iterative convergence. To generate the selective parameters at spatial window step $i$, the SPD input is first mapped to the Euclidean tangent space via the logarithmic map, yielding $S_i = \log(\mathcal{S}_{\text{all}}[i])$. The vectorized representation of $S_i$ is then processed by a lightweight Gated Recurrent Unit (GRU), whose output $O_i$ is passed through three linear layers to obtain the gating signal $\Delta_i$ and the selective operators. Formally, we compute

$$\Delta_i = \delta_{\max}\, \sigma\big(\text{softplus}(\text{fc\_delta}(O_i))\big), \quad B_i = \text{SR}\big(\text{fc\_B}(O_i)\big), \quad C_i = \text{SR}\big(\text{fc\_C}(O_i)\big),$$

where each $\text{fc}(\cdot)$ denotes a learnable linear map and $\text{SR}(\cdot)$ applies stiefel retraction to recover orthonormal operators on the manifold, guarantee that input injection and state projection remain consistent with SPD geometry. Unlike Mamba's selective parameterization, which operates purely in Euclidean space, our GRU evolves in the tangent space of the SPD manifold and is required to output geometry-compatible operators. Importantly, the log mapping is employed only for parameter generation; the input, hidden state, and output naturally remain within the SPD manifold, providing a unified foundation for stability and nonlinear coupling.

**Dynamic stability.** The core requirement of state iteration is dynamic stability. The evolution of the hidden state on the manifold is governed by the transition operator $A_i$, which enforces dynamic stability via spectral shrinkage. Formally,

$$A_i = \text{diag}\big(\exp\big(-\Delta_i \odot \lambda\big)\big), \tag{5}$$

where $\odot$ denotes element-wise multiplication. Here, $\Delta_i$ is an adaptive scalar step size for each sample and spatial window step , controlling the overall decay magnitude, while $\lambda_j$ is a learnable spectral parameter for the $j$-th state channel, constrained to be positive through the softplus function.

The hidden state is then updated by

$$U_{i+1} = A_i U_i A_i^T + B_i^T S_i B_i + \epsilon I, \tag{6}$$

where $B_i \in \mathbb{R}^{d \times d}$ is the selectively generated orthogonal mapping that injects input information, $S_i$ is the input at part $i$, and the perturbation term $\epsilon I$, $\epsilon = 1e^{-6}$ ensures positive definiteness. Since $A_i$ is diagonal and positive, the iterative update naturally preserves the SPD property of $U_i$ while coupling stability with selective dynamics under manifold constraints.

**Structure-preserving output mapping.** In the output phase of Riemannian SSM, we need to ensure two key goals simultaneously: on the one hand, the output result must be strictly maintained within the SPD manifold $\mathcal{M}$ to ensure consistency with the input and hidden states; On the other hand, the model should have robust expression capabilities across sequence of FC snapshots, thereby avoiding information degradation caused by relying solely on state updates. To this end, we introduce the output mechanism termed the *geodesic mixture* (*GeoMix*), as a foundational component designed to fuse model states and inputs on the SPD manifold in a geometry-aware manner. First, a pseudo-output $\tilde{Y}_t$ is obtained by projecting the predicted state $U_{i+1}$ via a learnable projection matrix $C_i$: $\tilde{Y}_i = C_i^\top U_{i+1} C_i$. Next, the *R-SSM* performs geometric fusion between the pseudo-output $\tilde{Y}_i$ and the input $S_i$ using the *GeoMix* operator:

$$Y_i = GeoMix(\tilde{Y}_i, S_i; \alpha), \quad \alpha \in [0, 1] \tag{7}$$

where $\alpha$ is a learnable fusion coefficient that balances the contributions of the predicted state and the input. Unlike conventional linear interpolation in Euclidean space, *GeoMix* performs geodesic interpolation under the Log-Euclidean metric, defined as:

$$GeoMix(A, B; \alpha) = \exp\left((1 - \alpha)\log(A) + \alpha\log(B)\right), \tag{8}$$

which ensures that the output $Y_t$ remains within the SPD manifold $\mathcal{M}$, preserving geometric consistency throughout the fusion process.

This design offers two key advantages. First, *GeoMix* provides an intrinsic, structure-preserving form of "addition" on the SPD manifold. Its convex combination in the logarithmic domain corresponds precisely to the shortest geodesic between $S_t$ and $\tilde{Y}_t$ under the Log-Euclidean metric, defined

as $d_{\text{LE}}(A, B) = \|\log A - \log B\|_F$. The interpolation curve $\gamma(\alpha) = \exp\big((1-\alpha)\log A + \alpha \log B\big)$ traces a constant-speed path of length $d_{\text{LE}}(A, B)$, making it the unique minimizing geodesic. This geodesic interpolation avoids the loss of positive-definiteness often caused by Euclidean linear combinations, thereby ensuring geometric consistency. Second, the *GeoMix* operation acts as a form of pseudo-residual connection: by explicitly incorporating the input signal $S_i$ into each output step, it mitigates gradient vanishing issues in long sequence modeling and facilitates stronger interactions between the input and latent state. Together, these properties allow the Riemannian SSM's output mechanism to maintain strict adherence to the SPD structure while supporting a robust and expressive information flow—akin to residual networks—thus enhancing both the stability and modeling capacity for sequence of FC snapshots on non-Euclidean domains.

### 2.4 HIERARCHICAL DIMENSION REDUCTION AND STACKED SELECTIVE STATE SPACE MECHANISM

To fully exploit the multi-scale geometric information encoded in the pyramid features and to mitigate redundancy in high-dimensional SPD inputs, we propose a hierarchical selective state space modeling framework consisting of two complementary branches, as illustrated in Fig. 1 (b). These two branches operate in parallel and are subsequently fused on the SPD manifold.

**The Stacked Path.** Given the concatenated *multi-scale feature sequence* $\mathcal{S}_{\text{all}}$, the first branch implements a *stacked* selective state space mechanism (*Path #1*). Specifically, $\mathcal{S}_{\text{all}}$ is first processed by a $\text{SSM}_1$ block, followed by a Euclidean self-attention module, and then by a *dSSM*$_1$ block. The Euclidean self-attention module enables adaptive integration of information across different spatial scales within the SPD sequence. By mapping each SPD matrix to the tangent (logarithmic) domain, it selectively weights contributions from various regions or scales in Euclidean space. The weighted representations are then mapped back to the SPD manifold via the exponential map, ensuring valid SPD matrices. This facilitates inter-scale feature interaction and enhances the expressiveness of the hierarchical SSM representations. Both $\text{SSM}_1$ and the self-attention module output SPD sequences of size $d \times d$, indexed across $(m + 1)$ spatial window steps. The final *dSSM*$_1$ module compresses these sequences into a single SPD matrix of size $l \times l$ by performing a weighted combination along the spatial axis. This 4-layers ($\text{SSM}_1$, self-attention module and 2-layers *dSSM*$_1$) stacked arrangement enables hierarchical spatial modeling and inter-scale interaction while preserving geometric consistency at each stage. The final output of this branch is denoted as $Y_{\text{stack}}$.

**The *Embedding Path*.** The second branch realizes a manifold-aware dimensionality reduction strategy (*Path #2*), referred to as the *embedding path*, which operates directly on the original high-dimensional SPD input $H \in \mathbb{R}^{n \times n}$ (Fig. 1 (c)). This path employs a progressive reduction mechanism using the *dSSM*$_2$ module, where each layer applies a *DimMap*-based projection to reduce the SPD matrix to a lower-dimensional subspace. Specifically, the process is formulated as:

$$H^{(l)} = \mathcal{R}_{d_l} \circ \mathcal{R}_{d_{l-1}} \circ \cdots \circ \mathcal{R}_{d_0}(H^{(0)}), \quad H^{(l)} \in \mathcal{M}_{++}^l, \tag{9}$$

where $H^{(0)}$ is the input SPD matrix, $\mathcal{R}_{d_i}(\cdot)$ denotes the manifold-preserving projection at the $i$-th layer, and $H^{(l)}$ is the final compact representation. This hierarchical compression reduces redundancy, retains Riemannian structure, and yields a low-dimensional SPD output $Y_{\text{down}}$. The number of compression layers is determined by the parameter *down_dim*. In the current implementation, 4-layers model is used.

**Manifold Fusion.** Finally, the outputs $Y_{\text{final}}$ from the stacked and *embedding paths*—$Y_{\text{stack}}$ and $Y_{\text{down}}$—are fused on the SPD manifold using the *GeoMix* operator. The resulting SPD matrix serves as the final compact representation, which is used for downstream classification tasks.

## 3 EXPERIMENT AND RESULTS

### 3.1 EXPERIMENTAL SETTING

**Datasets and baselines.** We employ six publicly available neuroimaging datasets (including 1847 subjects) for evaluation (detailed information is shown in Table 3, Appendix): ADNI (Mueller et al., 2005), OASIS (LaMontagne et al., 2019), PPMI (Marek et al., 2011), Taowu (Badea et al., 2017),

Neurocon (Badea et al., 2017), and Mātai (Xu et al., 2023). In ADNI, subjects are categorized into two groups: Category #1 includes normal cognition (CN), subjective memory complaint (SMC), and early mild cognitive impairment (EMCI), while Category #2 includes late mild cognitive impairment (LMCI) and Alzheimer's disease (AD). OASIS, Taowu, and Neurocon are binary-class datasets, where classes correspond to healthy controls versus patients with AD (OASIS) or PD (Taowu and Neurocon). PPMI consists of four groups: normal controls, scans without evidence of dopaminergic deficit (SWEDD), prodromal, and PD patients. Mātai is a longitudinal cohort evaluating the effects of contact sports, with two classes corresponding to pre-season and post-season scans. Regional mean time series are extracted using the AAL atlas (116 ROIs) (Tzourio-Mazoyer et al., 2002) for all datasets except OASIS, which uses the Destrieux atlas (160 ROIs) (Destrieux et al., 2010). The version of the data we use is based on preprocessed BOLD time series obtained through a standard neuroimaging pipeline (Xu et al., 2023; Dan et al., 2024).

We classify baseline methods into four categories: (1) Classical neural network and sequence modeling methods: MLP-Mixer (Tolstikhin et al., 2021), Transformer (Vaswani et al., 2017) and Mamba (Gu & Dao, 2023); (2) General graph neural networks: GCN (Kipf, 2016) and GIN (Xu et al., 2018); (3) Manifold learning models: SPDNet (Huang & Van Gool, 2017); (4) Brain network-specific models: BrainGNN (Li et al., 2021), NeuroGraph (Said et al., 2023), Contrast_Pool (Xu et al., 2024) and STAGIN (Kim et al., 2021). For the MLP-Mixer, GCN and GIN models, we uniformly set the hidden layer dimension to 256, and trained the models using the Adam optimizer with a learning rate of $5.0 \times 10^{-4}$ and weight decay of $1.0 \times 10^{-4}$. The remaining models only adapted the random seed, input dimensions, and number of categories to the brain states, while all other hyperparameters followed the defaults in the official code. For the Euclidean sequence baselines (Transformer, Mamba, MLP-Mixer), the input was the vectorized FC matrices derived from Pearson correlation of BOLD time series, following the standard practice in sequence modeling for FC-based representations. For graph-based methods, graphs are constructed from the FC matrices following the original implementations. For methods incompatible with FC inputs (e.g., BrainGNN, STAGIN), we use raw BOLD sequences with their native preprocessing.

**Implemental details.** To rigorously evaluate the classification performance of the proposed *Geo-Mamba* model, we employ cross-validation to ensure both the reliability and generalizability of results. For datasets with relatively small sample sizes (ADNI, PPMI, Mātai, Taowu, and Neurocon), we adopt a 10-fold cross-validation strategy to maximize the use of limited labeled data and enhance the robustness of statistical inference. For the larger OASIS dataset, a 5-fold cross-validation scheme is used to balance evaluation reliability with computational efficiency. We performed a grid search over the hyperparameter ranges defined in Table 4 (Appendix), and the reported Accuracy (Acc), Precision (Pre), and F1-score (F1) correspond to the best configuration found. To ensure the orthogonality and SPD constraints are properly maintained during optimization, we employ a manifold-based optimizer (Huang & Van Gool, 2017; Dan et al., 2022b;a) built on top of the Adam optimizer (Kingma & Ba, 2014). This wrapper integrates Riemannian gradient updates within Adam, allowing all StiefelParameters to remain on the SPD manifold throughout training. Thus, while Adam handles adaptive learning rates, the manifold wrapper guarantees that all matrix constraints are preserved. All experiments are performed on a Linux server equiped with dual Intel Xeon Platinum 8163 CPUs (2.50 GHz, 48 cores/96 threads), two NVIDIA RTX A6000 GPUs (48 GB each, CUDA 12.4), 32 GB RAM and 1 TB storage. Regarding the model parameters, running time and the corresponding comparative analysis, we have made further supplementary explanations in Table 5 (Appendix).

### 3.2 OVERALL PERFORMANCE

**Comparative analysis**. Table 1 reports the performance—Acc, Pre, and F1 score—across six neuroimaging datasets. Optimal results are bolded, second-best are underlined, and significance levels from paired t-tests are marked with ($* : p < 0.05$) and ($** : p < 0.001$). Overall, graph neural networks (GNNs; e.g., GCN, GIN, BrainGNN, NeuroGraph) outperform sequence models (e.g., MLP-Mixer, Transformer), underscoring the importance of brain network topology in classifying neurodegenerative disorders and motor-related phenotypes. Nonetheless, performance varies across datasets, reflecting both the strengths and limitations of existing approaches. Conventional GNNs like GCN and GIN offer stable performance but fall short in precision and F1 scores, highlighting architectural limitations. SPDNet performs well on large datasets (ADNI, OASIS) by modeling SPD matrices via Riemannian geometry but struggles with smaller, noisier datasets due to data

sparsity. Brain-specific models such as NeuroGraph and STAGIN leverage graph structure and temporal dynamics, performing well on PPMI but declining on more heterogeneous cohorts like ADNI. Contrast_Pool shows high variability, especially in small-sample scenarios, reflecting the limitations of contrastive learning in such settings. In contrast, *Geo-Mamba* achieves consistently strong results across all six benchmarks, outperforming or matching state-of-the-art baselines like Mamba, SPDNet, and STAGIN. It shows leading Accuracy and F1 scores on large-scale cohorts, robustness on small datasets, and competitive performance even in challenging settings like PPMI. These gains stem from its principled integration of manifold-aware representations and state-space modeling, capturing intrinsic geometry and spatial dynamics for robust neuroimaging analysis.

Table 1: Performance comparison on eleven methods across datasets.

| Method | Metric | ADNI | OASIS | PPMI | Taowu | Neurocon | Mātai |
|---|---|---|---|---|---|---|---|
| MLP-Mixer | Acc | 72.80 ± 9.00 ** | 87.51 ± 2.51 * | 66.05 ± 7.33 * | 87.50 ± 17.68 | 87.50 ± 17.68 | 81.67 ± 14.59 * |
| | Pre | 68.32 ± 11.49 * | 82.68 ± 4.33 | 64.90 ± 8.57 | 91.25 ± 18.45 | 84.79 ± 25.35 | 86.53 ± 12.42 |
| | F1 | 64.11 ± 10.80 ** | 81.99 ± 3.21 ** | 62.05 ± 7.58 ** | 88.10 ± 16.49 | 84.43 ± 22.90 | 81.48 ± 14.71 * |
| Transformer | Acc | 74.00 ± 7.60 ** | 87.84 ± 2.20 ** | 63.97 ± 9.32 * | 87.50 ± 17.68 | 87.50 ± 17.68 | 65.00 ± 14.59 ** |
| | Pre | 66.67 ± 12.96 * | 83.58 ± 3.83 | 57.80 ± 11.78 | 82.08 ± 26.92 | 87.92 ± 23.28 | 53.33 ± 26.94 ** |
| | F1 | 65.84 ± 8.60 ** | 83.48 ± 1.64 ** | 55.06 ± 12.30 ** | 83.52 ± 23.55 | 85.67 ± 22.00 | 54.88 ± 20.15 ** |
| GCN | Acc | 74.80 ± 7.55 ** | 87.67 ± 2.62 * | 66.00 ± 9.33 * | 85.00 ± 24.15 | 82.50 ± 16.87 * | 81.67 ± 16.57 |
| | Pre | 71.16 ± 11.19 * | 84.65 ± 3.01 | 62.81 ± 9.80 | **97.08 ± 6.23** | 84.58 ± 22.40 | 83.89 ± 21.91 |
| | F1 | 68.29 ± 8.76 ** | 83.33 ± 3.52 * | 57.05 ± 11.53 ** | 87.57 ± 19.76 | 80.33 ± 20.99 * | 79.35 ± 20.79 |
| GIN | Acc | 78.00 ± 7.36 | 87.43 ± 3.10 * | 67.00 ± 10.56 | 92.50 ± 12.08 | 95.00 ± 10.54 | 83.33 ± 17.57 |
| | Pre | 79.66 ± 6.81 | 81.69 ± 7.94 | 59.84 ± 14.76 | 95.83 ± 13.84 | 96.67 ± 7.03 | 88.44 ± 11.11 |
| | F1 | 73.00 ± 9.32 * | 82.18 ± 4.61 * | 60.30 ± 11.45 * | 93.57 ± 11.88 | 94.67 ± 11.25 | 81.22 ± 19.46 |
| SPDNet | Acc | 78.40 ± 9.08 | 89.14 ± 2.97 | 64.97 ± 9.99 * | 77.50 ± 18.45 * | 70.00 ± 19.72 * | 80.00 ± 20.49 * |
| | Pre | 79.87 ± 9.44 | 88.09 ± 3.33 | 58.31 ± 16.37 * | 95.83 ± 6.80 | 66.87 ± 31.28 * | 82.19 ± 22.96 * |
| | F1 | 74.08 ± 11.34 * | 86.66 ± 3.95 | 55.67 ± 13.14 * | 83.14 ± 13.29 * | 63.76 ± 25.70 * | 76.27 ± 24.62 * |
| BrainGNN | Acc | 79.20 ± 5.00 | 87.25 ± 1.42 | 67.18 ± 6.85 | 80.00 ± 10.00 * | 80.00 ± 18.71 * | 83.33 ± 10.54 |
| | Pre | 79.50 ± 10.20 | 76.15 ± 2.49 * | 61.49 ± 11.35 | 82.08 ± 14.29 | 73.33 ± 25.39 * | 81.64 ± 16.08 |
| | F1 | 75.12 ± 7.29 | 81.32 ± 2.04 * | 59.69 ± 8.80 * | 77.52 ± 11.97 * | 74.05 ± 23.05 * | 80.73 ± 13.15 |
| Contrast_Pool | Acc | 76.00 ± 6.93 * | 87.27 ± 2.72 * | 67.50 ± 9.16 | 82.50 ± 16.01 * | 92.50 ± 11.46 | 81.67 ± 17.40 |
| | Pre | 57.2 ± 19.84 * | 50.33 ± 14.27 * | 42.71 ± 19.82 * | 75.00 ± 28.32 * | 90.42 ± 18.82 | 74.75 ± 28.38 |
| | F1 | 56.19 ± 13.54 * | 48.07 ± 3.48 ** | 39.55 ± 16.68 ** | 73.90 ± 25.10 * | 88.95 ± 18.62 | 72.66 ± 25.79 * |
| NeuroGraph | Acc | 76.40 ± 7.41 * | 87.13 ± 3.00 | 69.08 ± 9.02 * | 87.50 ± 17.68 * | 82.50 ± 12.08 * | 81.67 ± 9.46 * |
| | Pre | 80.02 ± 5.90 | **88.86 ± 2.22** | **79.13 ± 3.07 ** | 89.58 ± 16.32 | 89.58 ± 9.00 * | 86.83 ± 17.36 |
| | F1 | 71.52 ± 8.82 * | 81.16 ± 4.29 * | 61.12 ± 11.43 ** | 85.19 ± 19.91 * | 80.76 ± 14.52 * | 76.74 ± 10.86 * |
| STAGIN | Acc | 77.60 ± 6.31 | 88.56 ± 0.78 | 69.66 ± 5.71 | 90.00 ± 12.91 | 92.50 ± 12.08 | 86.67 ± 10.54 |
| | Pre | 62.75 ± 38.97 | 83.21 ± 17.41 | 69.66 ± 5.71 | 90.00 ± 16.10 | 97.50 ± 7.91 | **90.83 ± 14.93** |
| | F1 | 62.71 ±± 27.06 * | 83.28 ± 10.13 * | 69.66 ± 5.71 | 90.67 ± 12.65 | 93.24 ± 11.82 | 81.24 ± 14.64 |
| Mamba | Acc | 79.20 ± 4.66 | 88.33 ± 1.99 * | 64.21 ± 3.94 | 92.50 ± 12.08 | 87.50 ± 12.50 * | 85.00 ± 15.72 |
| | Pre | 78.19 ± 8.88 | 85.83 ± 2.75 | 59.91 ± 11.65 | 94.37 ± 13.24 | 92.08 ± 8.00 * | 89.19 ± 10.17 |
| | F1 | 75.31 ± 5.49 | 84.72 ± 2.81 * | 59.91 ± 7.45 * | 92.67 ± 12.19 | 87.00 ± 13.03 * | 83.25 ± 17.37 |
| *Geo-Mamba* | Acc | **81.20 ± 6.21** | **89.39 ± 1.91** | **71.16 ± 9.12** | **97.50 ± 7.50** | **97.50 ± 7.50** | **91.67 ± 11.18** |
| | Pre | **81.10 ± 8.12** | 88.82 ± 2.32 | 64.81 ± 10.10 | 95.63 ± 13.13 | **98.33 ± 5.00** | 88.89 ± 17.60 |
| | F1 | **79.08 ± 7.43** | 87.10 ± 2.57 | 65.57 ± 10.17 | 96.43 ± 10.71 | 97.33 ± 8.00 | 89.48 ± 14.99 |

**Interpretable functional dynamics via learned feature representation.** To interpret the critical connection patterns associated with specific diseases, our analysis primarily relies on the matrices reconstructed via *Path #2*. This choice is motivated by the fact that reconstructing the fused low-dimensional representation back into the original SPD space requires a stable and geometrically coherent inverse projection. Since *Path #2* performs dimensionality reduction directly on the global SPD manifold, the resulting projection matrix is well suited for faithful reconstruction. In contrast, *Path #1* relies on local multi-scale aggregation and does not yield a mapping that supports accurate SPD recovery. We then reconstructed the low-dimensional representations obtained from the final stage of *Geo-Mamba* (i.e., $Y_{\text{final}}$) using the projection matrix from *Path #2*. Specifically, we apply the reconstruction formula $W_l Y_{\text{final}} W_l^\top$, where $W_l$ denotes the learned weighted matrix (obtained from $\mathcal{R}_{d_l}$) and $Y_{\text{final}}$ is the learned low-dimensional embedding. This operation restores the representation back to the original feature space ($n \times n$), allowing direct interpretation of disease-relevant connection patterns. After that, we average the resulting connectivity matrices within each diagnostic group to obtain a representative mean connectivity matrix. From this matrix, the top eleven strongest non–self-loop connections are identified and selected for brain network visualization.

As shown in Fig. 2, analysis across three Parkinson's disease (PD) datasets—Taowu, Neurocon, and PPMI—reveals that functional abnormalities are primarily localized within the Sensorimotor and Frontoparietal networks (Delgado-Alvarado et al., 2023; Tessitore et al., 2014), with additional involvement of the Default Mode and Cerebellar networks (Palmer et al., 2021; Ruppert et al., 2021; Boord et al., 2017). These consistent patterns suggest that disruptions in motor and higher-order control systems may underlie the hallmark motor and cognitive symptoms of PD. In contrast, for Alzheimer's disease (AD), the most prominent alterations are observed in the Ventral Attention and

Default Mode networks (Seoane et al., 2024; Chumin et al., 2021; Wang et al., 2019), with supplementary abnormalities in the Visual network (Han et al., 2024; Sheng et al., 2024). Regionally, impaired functional connectivity predominantly affects the Temporal, Frontal, and Insular lobes (Zhao et al., 2023; Tian et al., 2024), aligning with known trajectories of AD pathology. Moreover, in the Mātai dataset, *Geo-Mamba* identifies season-dependent reorganization of brain networks in athletes, with enhanced functional connectivity observed across the Frontoparietal, Visual, Sensorimotor, and Ventral Attention networks. These patterns likely reflect neuroplastic adaptations to intensive physical training. Together, these findings underscore the robustness of *Geo-Mamba* not only in detecting disease-related alterations, but also in capturing structured patterns of dysconnectivity within functional subnetworks. By operating at the whole-brain level and preserving intrinsic geometric and topological information, *Geo-Mamba* moves beyond isolated connection analysis to reveal coherent network-level reorganization.

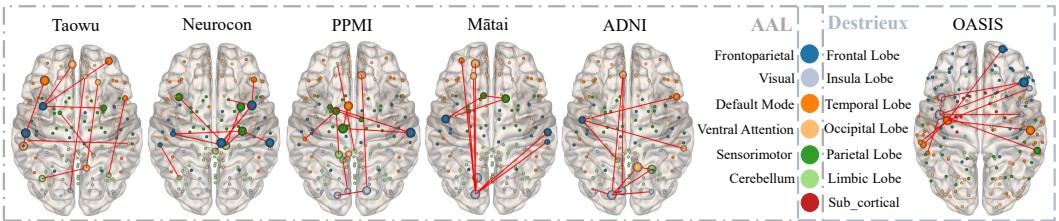

Figure 2: Each brain map illustrates the top 11 functional connections, highlighting the relationships among major brain regions. The node size reflects the ranking of the top connections, and red edges indicate the presence of a connection between two nodes. Note that the labels on the left and right sides of the legend correspond to the respective maps.

**Ablation study.** To assess the contribution of each component in *Geo-Mamba*, we perform comprehensive ablation studies across six neuroimaging datasets, as shown in Fig. 3. In each ablation setting, a key module was removed or replaced to evaluate its individual impact on disease diagnosis. The configurations include: (i) removing the local pyramid branch, retaining only the global *DimMap* path; (ii) removing the global path, relying solely on multi-scale local features; (iii) removing the *Path #2* downsampling branch, leaving only attention-based global compression; and (iv) bypassing the *Path #1*, which disables the initial manifold-aware spatial evolution and eliminating the sequence fusion under SSM and relying solely on the average spatial window step matrix for subsequent computations. The ablation study reveals that the contribution of each module in *Geo-Mamba* is closely tied to the complexity and scale of the datasets. On small-sample datasets such as Taowu and Neurocon, performance degradation from module removal is relatively minor, and in some cases, model performance remains stable. This suggests that when discriminative patterns are strong and localized, the remaining modules can partially compensate, and reduced model complexity may even mitigate overfitting. In contrast, on large-scale, feature-rich datasets such as ADNI and PPMI the removal of any critical path results in a substantial decline in performance. These datasets demand the joint modeling of fine-grained local patterns, long-range spatial dependencies, and dynamics—capabilities that are compromised when any core module is removed. The ablation results underscore the importance of *Geo-Mamba*'s full architecture for robust and scalable neuroimaging analysis. To further verify the independent contributions of the three key components in *Path #1* - $SSM_1$, self-attention, and $dSSM_1$ - we additionally designed ablation experiments v - vii, which respectively correspond to the removal of the above three modules. The relevant results and analyses have been presented in Table 6 (Appendix).

To assess the relative contribution of the two paths, we examined the GeoMix weights $\alpha$ after training (*Path #1* weights — Taowu 0.5271, Neurocon 0.6329, PPMI 0.6958, Matai 0.5191, ADNI 0.7247, OASIS 0.6807). Both *Path #1* and *Path #2* contribute meaningfully to the final representation, with *Path #1* generally having a slightly higher influence, reflecting the dominant role of local–global multi-scale features. The non-negligible weights assigned to *Path #2* indicate that global SPD manifold information provides complementary insights. Overall, these results confirm that the GeoMix layer effectively balances local and global features to generate robust fused embeddings across datasets.

These findings not only demonstrate the necessity of each module for optimal performance on complex datasets, but also reveal their distinct and complementary roles within the overall architecture. Specifically, the local pyramid branch is indispensable for capturing multi-scale regional alterations; without it, fine-grained variations in connectivity are lost, particularly harming datasets like PPMI

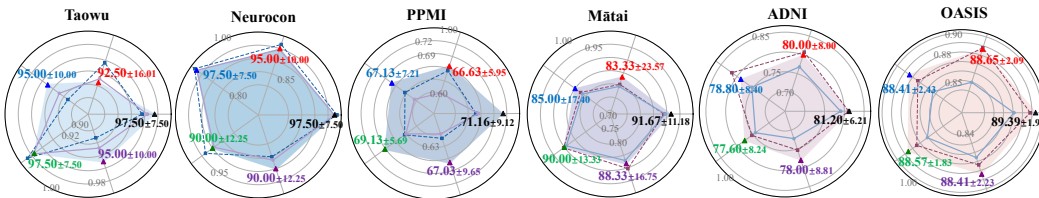

Figure 3: Radar plots of ablation results. Lines represent F1-scores, dashed lines indicate precision, and shaded areas correspond to accuracy. Black fonts denote the full *Geo-Mamba* model, while red, blue, green, and purple fonts indicate ablation settings (i–iv), respectively. The radial axes are selectively rescaled for better visualization, and accuracy values are annotated as mean ± standard deviation.

where subtle local anomalies are key to stratification. The global integration path plays a different but equally crucial role by aggregating distributed weak signals across the brain into coherent representations, which explains the marked decline on ADNI and Mātai when this path is removed. Meanwhile, the manifold-aware spatial evolution in *Path #1* enables the model to retain dynamic consistency along the sequence dimension; bypassing it reduces the representation to static averages, which fail to capture time-dependent pathological signatures, especially in heterogeneous cohorts. Although the absolute performance drop after removing *Path #1* appears moderate, the geometric constraints provide additional benefits beyond accuracy—namely preserving SPD validity, stabilizing the underlying dynamics, and yielding smoother manifold-aware evolutions. Finally, the compression mechanism in the downsampling path stabilizes the feature space by providing compact yet geometry-preserving representations, and its removal leads to noisier and less discriminative embeddings, which undermines robustness on complex datasets. In summary, the superior performance of *Geo-Mamba* arises from the synergistic integration of its components, each essential for ensuring robustness and adaptability across diverse neuroimaging datasets.

Table 2: Comparative analysis of dynamic feature construction methods.

| Method | Taowu | | | OASIS | | |
|---|---|---|---|---|---|---|
| | Acc | Pre | F1 | Acc | Pre | F1 |
| Sliding | $95.00 \pm 10.00$ | $93.96 \pm 13.52$ | $93.76 \pm 12.64$ | $88.97 \pm 1.77$ | $88.58 \pm 1.09$ | $85.83 \pm 2.55$ |
| Ours | $\mathbf{97.50 \pm 7.50}$ | $\mathbf{95.63 \pm 13.13}$ | $\mathbf{97.50 \pm 7.50}$ | $\mathbf{89.39 \pm 1.91}$ | $\mathbf{88.82 \pm 2.32}$ | $\mathbf{89.39 \pm 1.91}$ |

**Discussion.** In conventional fMRI analysis, dynamic features are typically extracted using the sliding window method (Preti et al., 2017; Cavanna et al., 2018; Shunkai et al., 2023; Dan et al., 2022a). In contrast, our model constructs a *multi-scale feature sequence* (Eq. 4) by aggregating multi-scale spatial features and global context representations, without relying on explicit temporal slicing. To compare the two approaches, we replaced the pyramid module with a sliding window module while keeping all other components fixed. As shown in Table 2, our pyramid-based design consistently outperforms the sliding window method (window size = 60, number of windows = 5) on both Taowu (small sample) and OASIS (large-scale) datasets—showing especially notable gains under limited data conditions. These improvements stem from several advantages: (i) the tailored *multi-scale feature sequence* avoids the instability of window size selection and leverages the full temporal signal; (ii) Log-Euclidean attention pooling preserves SPD geometry, enhancing discriminability; and (iii) the sequence structure encodes a semantically aligned progression from local to global representations, offering more stable and interpretable inputs for dynamic modeling. Together, this geometry-aware spatial encoding proves more robust and effective than conventional temporal slicing, particularly in data-limited or other challenging scenarios.

## 4 CONCLUSION

In this work, we propose *Geo-Mamba*, a geometric state space model for dynamic fMRI analysis on Riemannian manifolds. By extending selective SSMs to SPD manifolds and integrating pyramid feature extraction with hierarchical dimensionality reduction, *Geo-Mamba* enables multi-scale, non-linear, and geometry-preserving modeling of brain functional dynamics. Experiments on multiple neuroimaging datasets show that *Geo-Mamba* robustly captures spatiotemporal brain patterns, with its core modules jointly ensuring manifold consistency, spatial modeling capacity, and representational robustness. Overall, *Geo-Mamba* establishes a new paradigm for geometric deep learning in neuroimaging, with potential extensions to other non-Euclidean time series such as EEG and motion capture.

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

# A APPENDIX

**Use of Large Language Models (LLM)**. In the preparation of this manuscript, large language models were employed to aid and polish the writing. These models provided assistance in refining the clarity and coherence of the text, ensuring that the content adheres to high standards of academic writing.

**Availability of Code**. To facilitate reproducibility and further research, the code used in this study is publicly available at Anonymous GitHub. The repository includes all necessary scripts and documentation for replicating or extending our work.

**Ethical Considerations and Data Usage**. In adherence to the ICLR Code of Ethics, our study exclusively employs publicly available neuroimaging datasets that were collected and released in accordance with established ethical standards. These datasets are accessible for application and download from their respective websites, and their use in scientific research inherently complies with ethical, transparent, and responsible practices. Although our work involves human subjects through these datasets, no additional ethical approval is required since the datasets have already undergone rigorous ethical review processes prior to public release. We have also ensured that our use of these datasets complies with the ethical guidelines provided by the respective data repositories, which include considerations for privacy, confidentiality, and the responsible use of data.

**Conflict of Interest**. We declare that there are no conflicts of interest regarding the use of these datasets. Our research is conducted solely with the aim of contributing to scientific knowledge and public welfare, and we have taken steps to ensure that our work is free from any undue influence or bias. Furthermore, all authors confirm that they have no personal, financial, or professional conflicts of interest related to this study.

**Supplementary Tables**. For completeness, this appendix compiles the supplementary tables and corresponding analyses referenced throughout the paper.

Tables 3, 4 and 5 serve as supplementary tables for Section 3.1, presenting the detailed dataset statistics, the general hyperparameter configurations of the *Geo-Mamba* and specific information about model parameters and running time. For clarity, training and inference times were measured over one full epoch using representative datasets: Taowu for ROI-116 (batch size 4) and OASIS for ROI-160 (batch size 8), each including all train/test samples.

Table 3: The summarization of datasets.

| Dataset | # Subjects | # Classes | # Mean Time Points | # ROIs |
|---------|-----------|-----------|--------------------|--------|
| ADNI | 250 | 2 | 177 | 116 |
| OASIS | 1,247 | 2 | 390 | 160 |
| PPMI | 209 | 4 | 198 | 116 |
| Taowu | 40 | 2 | 239 | 116 |
| Neurocon | 41 | 2 | 137 | 116 |
| Mātai | 60 | 2 | 200 | 116 |

Table 6 supplements the ablation experimental results in Section 3.2. ablation I-IV corresponds to the four groups of control experiments reported in the main text; ablation V-VII respectively examined the independent contributions of the three key components ($SSM_1$, self-attention module and $dSSM_1$) in *Path #1*.

It can be observed from the results that the removal of $SSM_1$ (ablation v) has a relatively small impact on datasets with smaller data volumes such as Taowu and Neurocon, indicating the complementary effects of other modules. However, the performance was significantly reduced on ADNI, Mātai and PPMI. This phenomenon reflects that in tasks with more ambiguous category boundaries and more subtle differences in spatial patterns, the fundamental spatial dynamic modeling provided by $SSM_1$ still plays a key role in distinguishing different populations. The removal of the self-attention module (ablation vi) did not lead to the most significant performance degradation, but its effect showed an uneven characteristic among different datasets. Especially on datasets with complex spatial structure differences such as PPMI, the model performance has deteriorated to a certain

Table 4: The summarization of general hyperparameters for *Geo-Mamba*.

| Parameter | Value |
|---|---|
| learning rate | $1.0 \times 10^{-3}, 2.0 \times 10^{-3}\ 3.0 \times 10^{-3}, 4.0 \times 10^{-3}, 5.0 \times 10^{-3}$ |
| weight decay | $1.0 \times 10^{-4}, 2.0 \times 10^{-4}, 3.0 \times 10^{-4}, 4.0 \times 10^{-4}, 5.0 \times 10^{-4}$ |
| window_sizes | [16, 32, 64, 96] |
| down_dimension | [96, 64, 32, 16] |
| d_state | 32 |
| epoch | 100 |
| seed | 1114 |
| batch size | 4 (PPMI, Neurocon, Taowu, Matai), 5 (ADNI), 8 (OASIS) |

Table 5: Model parameters and computational cost comparison.

| Method | Dataset | Parameter | train time | test time |
|---|---|---|---|---|
| SPDNet | Taowu | 10186 | 0.1746/s | 0.0153/s |
| | OASIS | 402882 | 25.6739/s | 6.1364/s |
| Mamba | Taowu | 13942906 | 0.6222/s | 0.0215/s |
| | OASIS | 26458754 | 44.3853/s | 6.3147/s |
| Geo-Mamba | Taowu | 3923850 | 2.2933/s | 0.1426/s |
| | OASIS | 3929482 | 72.3225/s | 12.5951/s |

extent, indicating that this module plays an auxiliary role in adjusting and fusing multi-scale spatial information. Although its contribution is not as significant as that of $dSSM_1$, the absence of self-attention still weakens the flexibility and sensitivity of the model when dealing with cross-scale spatial dependencies, reflecting its steady-state effect on the overall representational ability. The removal of $dSSM_1$ (ablation vii) led to a significant decrease on all datasets, demonstrating its core role in modeling spatial structure evolution and improving overall discriminative power.

Overall, the importance of the three components shows hierarchical differences: the contribution of $dSSM_1$ is the most stable and significant, followed by the necessary support of $SSM_1$ for complex datasets, while the self-attention module provides beneficial but non-decisive multi-scale spatial information regulation capabilities.

Table 6: Performance comparison on ablation study across datasets.

| Method | Metric | ADNI | OASIS | PPMI | Taowu | Neurocon | Mātai |
|---|---|---|---|---|---|---|---|
| ablation i | Acc | $80.00 \pm 8.00$ | $88.65 \pm 2.09$ | $66.63 \pm 5.95$ | $92.50 \pm 16.01$ | $95.00 \pm 10.00$ | $83.33 \pm 23.57$ |
| | Pre | $80.74 \pm 8.67$ | $88.81 \pm 2.40$ | $65.50 \pm 6.48$ | $95.63 \pm 13.13$ | $96.67 \pm 6.67$ | $80.36 \pm 31.98$ |
| | F1 | $76.80 \pm 9.75$ | $84.97 \pm 2.71$ | $60.03 \pm 6.16$ | $93.10 \pm 13.82$ | $94.67 \pm 10.67$ | $79.54 \pm 29.83$ |
| ablation ii | Acc | $78.80 \pm 8.40$ | $88.41 \pm 2.43$ | $67.13 \pm 7.21$ | $95.00 \pm 10.00$ | $97.50 \pm 7.50$ | $85.00 \pm 17.40$ |
| | Pre | $81.59 \pm 7.78$ | $87.33 \pm 1.80$ | $63.36 \pm 10.29$ | $91.25 \pm 17.50$ | $98.33 \pm 5.00$ | $82.44 \pm 25.18$ |
| | F1 | $75.36 \pm 9.38$ | $85.37 \pm 3.79$ | $60.63 \pm 9.08$ | $92.86 \pm 14.29$ | $97.33 \pm 8.00$ | $81.49 \pm 22.78$ |
| ablation iii | Acc | $77.60 \pm 8.24$ | $88.57 \pm 1.83$ | $69.13 \pm 5.69$ | $97.50 \pm 7.50$ | $90.00 \pm 12.25$ | $90.00 \pm 13.33$ |
| | Pre | $75.36 \pm 10.01$ | $87.46 \pm 2.72$ | $63.45 \pm 9.02$ | $98.75 \pm 3.75$ | $93.33 \pm 8.16$ | $90.36 \pm 12.93$ |
| | F1 | $74.24 \pm 8.92$ | $86.08 \pm 2.14$ | $63.08 \pm 6.22$ | $97.67 \pm 7.00$ | $89.33 \pm 13.06$ | $88.78 \pm 14.63$ |
| ablation iv | Acc | $78.00 \pm 8.81$ | $88.41 \pm 2.23$ | $67.03 \pm 9.65$ | $95.00 \pm 10.00$ | $90.00 \pm 12.25$ | $88.33 \pm 16.57$ |
| | Pre | $75.31 \pm 13.75$ | $87.39 \pm 2.46$ | $61.08 \pm 11.12$ | $91.25 \pm 17.50$ | $85.21 \pm 19.58$ | $90.36 \pm 12.93$ |
| | F1 | $72.27 \pm 11.61$ | $86.57 \pm 2.57$ | $59.94 \pm 11.29$ | $92.86 \pm 14.29$ | $86.62 \pm 16.57$ | $87.10 \pm 18.10$ |
| ablation v | Acc | $77.60 \pm 8.24$ | $88.81 \pm 2.04$ | $70.16 \pm 8.50$ | $97.50 \pm 7.50$ | $97.50 \pm 7.50$ | $88.33 \pm 7.64$ |
| | Pre | $74.59 \pm 13.97$ | $87.60 \pm 2.68$ | $64.07 \pm 12.47$ | $95.63 \pm 13.13$ | $98.33 \pm 5.00$ | $89.36 \pm 8.72$ |
| | F1 | $73.61 \pm 11.35$ | $86.41 \pm 2.07$ | $63.76 \pm 10.91$ | $96.43 \pm 10.71$ | $97.33 \pm 8.00$ | $87.15 \pm 8.66$ |
| ablation vi | Acc | $80.40 \pm 7.47$ | $89.22 \pm 1.94$ | $67.63 \pm 5.92$ | $97.50 \pm 7.50$ | $97.50 \pm 7.50$ | $90.00 \pm 11.06$ |
| | Pre | $79.79 \pm 7.40$ | $88.16 \pm 2.73$ | $63.46 \pm 11.26$ | $95.63 \pm 13.13$ | $98.33 \pm 5.00$ | $89.86 \pm 12.19$ |
| | F1 | $77.02 \pm 9.34$ | $87.05 \pm 1.55$ | $59.34 \pm 7.72$ | $96.43 \pm 10.71$ | $97.33 \pm 8.00$ | $89.10 \pm 11.80$ |
| ablation vii | Acc | $77.20 \pm 9.30$ | $88.89 \pm 1.87$ | $68.61 \pm 6.79$ | $90.00 \pm 16.58$ | $90.00 \pm 16.58$ | $90.00 \pm 13.33$ |
| | Pre | $72.87 \pm 10.30$ | $85.69 \pm 4.09$ | $60.46 \pm 11.38$ | $88.96 \pm 18.64$ | $83.75 \pm 26.10$ | $92.03 \pm 10.64$ |
| | F1 | $71.96 \pm 10.18$ | $86.72 \pm 2.48$ | $60.92 \pm 8.53$ | $88.76 \pm 17.95$ | $86.19 \pm 22.56$ | $88.78 \pm 14.63$ |

