# OpenReview forum: "Geo-Mamba: Dual-Path Riemannian State-Space Models for Functional Dynamics"
_ICLR.cc/2026/Conference — Submitted to ICLR 2026_

### Official Review · Reviewer_Ai35 · 2025-10-24

**Soundness:** 2
**Presentation:** 3
**Contribution:** 3
**Rating:** 4
**Confidence:** 5

**Summary:**

This paper proposes an extension of the SPDNet architecture to the recently emerging Mamba state-space modeling framework, referred to as Geo-Mamba. The method aims to model brain functional connectivity matrices on the SPD manifold while leveraging Mamba’s efficient sequence modeling capacity. The authors conduct experiments on six public fMRI datasets and report consistent performance gains, suggesting that the proposed model effectively captures multi-scale geometric and temporal dependencies.

While the idea of integrating geometric deep learning on SPD manifolds with structured state-space models is interesting and timely, the paper lacks clarity in several conceptual, mathematical, and experimental aspects, which significantly weaken its overall rigor and reproducibility.

**Strengths:**

The paper addresses a clear architectural gap by bridging SPD-based geometric representation learning and the latest Mamba sequence model, combining two active but previously disconnected research domains (geometric deep learning and state-space modeling).

The motivation to extend SPDNet to temporal modeling via Mamba is intuitively appealing and potentially impactful for neural signal and fMRI analysis, where both geometric structure and dynamic dependencies are crucial.

The empirical study across multiple datasets demonstrates that the proposed approach can generalize across diverse fMRI sources, suggesting practical relevance if rigorously validated.

**Weaknesses:**

The paper does not clearly articulate why the SPD manifold is the most appropriate representation for fMRI data in this context. What type of statistical or functional information is being preserved by the SPD structure? Why is the log-Euclidean metric chosen over alternatives such as AIRM or Stein divergence? Without such justification, the methodological grounding feels ad hoc.

The implementation (as observed in MambaSPD_Attn_then_SSM) employs Riemannian Batch Normalization, which is derived for the AIRM metric, while the paper consistently claims to use the log-Euclidean metric. Mixing these two incompatible metrics within the same model undermines mathematical consistency and raises questions about the validity of the results.

Experimental design lacks important information: the number of subjects used in the experiments does not correspond to any known public version. The category definitions (e.g., diagnostic groups) and cross-validation protocols (5-fold vs. 10-fold) are inconsistent across datasets without explanation. Preprocessing steps, such as confound regression, normalization, or atlas selection, are insufficiently described, making reproducibility difficult, even with access to the code.

It is not clear how the SPD matrices are represented within the Mamba pipeline in your baseline comparison. From the description, the “Geo-Mamba” variant seems to apply a logarithmic mapping only at the final stage. Conceptually, this should yield behavior similar to standard Mamba, yet large performance differences are reported. Can you explain why?

**Questions:**

1. Could you elaborate on the motivation for using SPD representations in this task? What specific information from fMRI connectivity are you preserving that would be lost in a Euclidean embedding?

2. How do you reconcile the use of Riemannian Batch Normalization (AIRM-based) with your log-Euclidean metric formulation?

3. Please clarify the exact dataset versions, subject selection criteria, ROI definitions, and cross-validation strategies.

4. How are SPD matrices encoded as tensors in the Mamba architecture? Are manifold operations actually preserved throughout the sequence modeling pipeline? What explains the large performance differences between Geo-Mamba and vanilla Mamba across datasets if the only modification is a logarithmic mapping?

5. Figure 2 appears to show extremely sparse connectivity patterns. Was any thresholding or sparsification applied to produce the visualizations in Figure 2?

---

> ### Author Response · Authors · 2025-11-21
> **Respone to Reviewer Ai35**
>
> ### Thank you for acknowledging the contributions of our work. We are thrilled and grateful for your insightful feedback, which has significantly contributed to enhancing the quality of our manuscript. In the following responses, **W** and **Q** represent Weaknesses and Questions, and **A** represents the corresponding answer.
>
> **W1, Q1**. The motivation for using SPD was not clarified.
>
> **A1**:  The adoption of symmetric positive definite (SPD) matrix representation in this task is not an engineering choice but is based on considerations of the mathematical structure and biological implications of functional connectivity (FC) in fMRI. We have included a more detailed explanation of the above-mentioned motivation in the revised draft and cited the corresponding mathematical measures and experimental comparisons to support this design choice (**please refer to lines 38 to 48 on Page 1** ).
>
> **W2, Q2**. How do you reconcile the use of Riemannian Batch Normalization (AIRM-based) with your log-Euclidean metric formulation?
>
> **A2**:  Thank you for pointing this out. We realize that the name Riemannian Batch Norm SPD may have caused unnecessary confusion. To clarify, all transformation steps in our implementation are consistently based on the Log–Euclidean metric, including the normalization layer.  Although the term “Riemannian” was used in the module name, the actual computations do not rely on AIRM; they operate entirely in the log-domain following the Log–Euclidean framework. We have updated the code function name (RiemannianBatchNormSPD to LogExpBatchNormSPD) to clarify this point and avoid the impression that two different Riemannian metrics were mixed  (**please refer to lines 762 to 764 on Page 15 (Appendix)** ).
>
> **W3, Q3**. Please clarify the exact dataset versions, subject selection criteria, ROI definitions, and cross-validation strategies.
>
> **A3**:  Thank you for the reviewer’s question. Our data are sourced from two prior works: NIPS 2022 (10.48550/arXiv.2211.12421) and ICML 2024 (10.48550/arXiv.2405.16357). The ROI definitions and preprocessing pipelines are fully described in these original works. In our experiments, we directly used their preprocessed data (**please refer to lines 333 to 335 on Page 7**).
>
> Regarding cross-validation, to maximize the utility of smaller datasets such as Taowu and Neurocon, we employed 10-fold cross-validation. For larger datasets like OASIS, we adopted 5-fold cross-validation. This strategy ensures a fair and robust evaluation while respecting the size and characteristics of each dataset. We have made it clear in the manuscript (**please refer to lines 352 to 355 on Page 7** ).
>
> **W4, Q4**.  How are SPD matrices encoded as tensors in the Mamba architecture?  Are manifold operations actually preserved throughout the sequence modeling pipeline?  What explains the large performance differences between Geo-Mamba and vanilla Mamba across datasets if the only modification is a logarithmic mapping?
>
> **A4**:  Thank you for the reviewer’s question.   For the Euclidean sequence baselines (Mamba), the input was the vectorized FC matrices derived from Pearson-correlation of BOLD time series, following the standard practice in sequence modeling for FC-based representations.  The original Mamba model parameters are initialized following the publicly available code, with only a final linear classification layer added (similar to Fig. 1(b), classifier).
>
> Riemannian operations are preserved throughout the sequence modeling process. For SPD parameters, we employ a manifold-aware Adam optimizer to ensure updates respect the manifold constraints (**please refer to lines 357 to 362 on Page 7** ). Input data, together with manifold parameters and log/exponential mappings, maintain the symmetric positive definite property throughout training.
>
> The performance difference between Geo-Mamba and original Mamba is not solely due to the log mapping. Geo-Mamba is a geometric variant of Mamba on the SPD manifold, incorporating two complementary paths:
> - Stack path: performs hierarchical sptial modeling by aggregating multi-scale pyramid features, capturing both local and global dependencies.
> - Reduction path: applies progressive geometric-aware dimensionality reduction directly on the manifold, mitigating redundancy in high-dimensional SPD inputs while respecting Riemannian constraints.
>
> The outputs of these two paths are fused via a custom GeoMix operator, producing compact and discriminative SPD representations. This dual-path design, combined with manifold-preserving operations, explains the notable performance improvements of Geo-Mamba over the original Mamba across different datasets.

---

> > ### Author Response · Authors · 2025-11-21
> >
> > **Q5**.  Figure 2 appears to show extremely sparse connectivity patterns. Was any thresholding or sparsification applied to produce the visualizations in Figure 2?
> >
> > **A5**:  Thank you for the question. As described in the manuscript  (**please refer to lines 424 to 425 on Page 8** ), Figure 2 visualizes the top 11 strongest non-self-loop connections reconstructed from Geo-Mamba’s final-stage representation. This selection is intended to highlight disease-relevant connectivity patterns while keeping the figure clear and interpretable, and does not involve arbitrary thresholding beyond what is already described.

---

> > ### Comment · Reviewer_Ai35 · 2025-11-21
> >
> > A1 Line 40-42 in Page 1: “This geometric structure often leads to the distortion of statistical relationships in Euclidean embeddings due to the disruption of positive definiteness, making it difficult to preserve the overall pattern of brain region interactions and time-varying trajectories (You & Park (2021)).”
> >
> > This sentence reads very ambiguously. The distortion arises from ignoring the underlying geometry and applying inappropriate Euclidean operations, not from the manifold geometry itself. The loss of positive definiteness is a consequence of using incorrect Euclidean operations, not the mechanism that causes the distortion. It is also unclear why time-varying trajectories are mentioned in this context. Finally, the phrase “Euclidean embedding” is too vague and does not specify which operations (such as vectorization, PCA, or linear projections) fail to respect the Riemannian structure.
> >
> > A2: From my understanding, this approach appears to be the standard batch normalization applied in Euclidean space. Could you elaborate on the reasoning behind the name 'LogExpBatchNormSPD'? I'd like to understand how it specifically relates to the SPD manifold.
> >
> > A4: Why does performance improve so much? You skipped the main point of my question. Your explanation mentions “stack path”, “reduction path”, and “GeoMix”, but these descriptions are too vague. Please explain in concrete terms which part of Geo-Mamba (log map or architectural changes) causes the performance difference.

---

> ### Author Response · Authors · 2025-11-22
>
> Dear Reviewer,
>
> Thank you for your insightful comments. We deeply appreciate your perspective, as it has helped us reflect further on this important aspect of our work.
>
> **Modifications to A1**: We fully agree that the distortion arises not from the manifold geometry itself, but from inappropriate Euclidean operations (such as vectorization, PCA, or linear projections) that ignore the underlying Riemannian structure of SPD matrices. We have revised the description accordingly to clarify the mechanism (**please refer to lines 40 to 43 on Page 1** ). The updated version is shown below, with the modified text highlighted in bold.
>
> The overall structure of the FC matrix not only reflects pairwise correlation but also the covariance structure and dynamic stability at the network level. Therefore, it is more naturally regarded as a family of SPD matrices, **and distortions can arise when Euclidean operations (e.g., naive batch averaging , interpolation and  PCA) are applied without respecting the Riemannian geometry, breaking positive definiteness and disrupting the overall pattern of functional interactions**.
>
> By the way, extensive studies have analyzed SPD matrices as holistic objects on the manifold, **in order to overcome the limitations caused by directly using Euclidean metrics**, and we are happy to provide additional references to **provide a more comprehensive context [1-6]**.
>
>
> [1] 10.1007/s11263-005-3222-z
>
> [2] 10.1137/050637996
>
> [3] 10.48550/arXiv.1608.04233
>
> [4] 10.1016/j.neuroimage.2020.117464
>
> [5] 10.1002/mrm.20965
>
> [6] 10.1137/18M1221084
>
> **Supplement to A2**: The meaning of the name "LogExpBatchNormSPD" is: First, perform matrix logarithmic (Log) mapping on a Batch of SPD matrices, apply Batch Normalization similar to the standard on the logarithmic space (i.e., Euclidean space), and then send the results back to the SPD manifold through matrix exponential (Exp) mapping. Therefore, it is named "Log-Exp (log-exponential) + BatchNorm (Batch Normalization) + SPD (Symmetric Positive Definite Matrix)".
>
> The reason why we adopt log-Exp normalization is that the nonlinear structure of the SPD manifold makes the addition, subtraction and scaling operations relied upon by the standard BatchNorm ineffective on the manifold. Log-euclidean geometry provides a strictly reversible and geometrically consistent linearization framework, enabling SPD matrices to perform batch statistical calculations in the logarithmic domain in a Euclidean manner (**please refer to lines 131 to 155 on Page 3**). In contrast, directly implementing normalization on a manifold (for example, by means of parallel transport) requires repeated high-order matrix power operations, and the computational cost is unbearable. However, not performing normalization will lead to batch-to-batch distribution drift and unstable gradients, thereby seriously affecting the trainability of deep networks. Meanwhile, normalizing only in the eigenvalue or eigenvector domain will disrupt the feature structure and bring about problems of non-differentiability and numerical instability. Therefore, under the premise of ensuring the SPD structure, differentiability and training stability, Log-Exp is the most reasonable and feasible choice for manifold normalization.
>
> **In short**, our "LogExpBatchNormSPD" does not simply blindly normalize the SPD matrix in Euclidean space, but is based on the Log-Euclidean metric. This process is geometrically meaningful (maintaining the SPD properties and being compatible with the Log-Euclidean Riemannian metric), while being computationally efficient, differentiable and easy to integrate into existing training processes in implementation.

---

> > ### Comment · Reviewer_Ai35 · 2025-11-22
> >
> > Most of the papers you provided are stuck on technical details about SPD manifolds and Riemannian metrics. They completely ignore the real question: Why do we even need fMRI functional connectivity for this task? I’m looking for a neuroimaging-based rationale, not more math. I’ve seen many papers going down this path, but most are vague. Honestly, I’m starting to question whether this entire line of work is solid to begin with.
> >
> > Btw, bringing in new AI models with minor tweaks doesn’t address the real problem. Many neural network architectures may appear sophisticated, but often fall apart under scrutiny. This is not a solid research direction. I suggest you strip the problem down to its essentials so I can quickly judge whether such complicated designs are even necessary.

---

> ### Author Response · Authors · 2025-11-22
>
> **Supplement to A4**: We thank for the detailed feedback. In the revised manuscript, we added a comprehensive ablation study, which directly isolates the architectural factors responsible for the observed improvements(**please refer to lines 471 to 474 on Page 9**).Using the Matai dataset as an example:
>
> - Ablation-i (removing the local pyramid branch) and Ablation-ii (removing the global path) lead to a substantial degradation across all metrics, with performance dropping to a level close to Mamba. This indicates that the hierarchical local–global feature extraction module is the primary source of the improvements.
>
> - Ablation-iii (removing Path #2) and Ablation-iv (bypassing Path #1) result in only minor performance decreases, indicating that both paths effectively leverage the extracted features and, due to their complementary roles, the overall performance is largely preserved.
>
> - Ablation-v/vi/vii, which remove individual components within Path #1, also degrade performance, confirming that each submodule contributes meaningfully and that the gains arise from their joint design rather than any single operation.
>
> **Overall**, the performance improvement on the Matai dataset is driven by both the proposed feature extraction module, which captures key structural information, and the dual-path mechanism, which effectively leverages these extracted features; the two components work in parallel, and similar trends are observed on other datasets.
>
> | Method      | Metric | ADNI             | OASIS            | PPMI             | Taowu            | Neurocon         | Mātai            |
> |-------------|--------|------------------|------------------|------------------|------------------|------------------|------------------|
> | ablation i  | Acc    | 80.00 ± 8.00     | 88.65 ± 2.09     | 66.63 ± 5.95     | 92.50 ± 16.01    | 95.00 ± 10.00    | 83.33 ± 23.57    |
> | ablation ii | Acc    | 78.80 ± 8.40     | 88.41 ± 2.43     | 67.13 ± 7.21     | 95.00 ± 10.00    | 97.50 ± 7.50     | 85.00 ± 17.40    |
> | ablation iii| Acc    | 77.60 ± 8.24     | 88.57 ± 1.83     | 69.13 ± 5.69     | 97.50 ± 7.50     | 90.00 ± 12.25    | 90.00 ± 13.33    |
> | ablation iv | Acc    | 78.00 ± 8.81     | 88.41 ± 2.23     | 67.03 ± 9.65     | 95.00 ± 10.00    | 90.00 ± 12.25    | 88.33 ± 16.57    |
> | ablation v  | Acc    | 77.60 ± 8.24     | 88.81 ± 2.04     | 70.16 ± 8.50     | 97.50 ± 7.50     | 97.50 ± 7.50     | 88.33 ± 7.64     |
> | ablation vi | Acc    | 80.40 ± 7.47     | 89.22 ± 1.94     | 67.63 ± 5.92     | 97.50 ± 7.50     | 97.50 ± 7.50     | 90.00 ± 11.06    |
> | ablation vii| Acc    | 77.20 ± 9.30     | 88.89 ± 1.87     | 68.61 ± 6.79     | 90.00 ± 16.58    | 90.00 ± 16.58    | 90.00 ± 13.33    |

---

> ### Author Response · Authors · 2025-11-23
> **Academic discussions with Reviewer Ai35**
>
> Dear Reviewer Ai35,
>
> Thank you for your insightful response, **we enjoy academic discussions like this**.
>
> **Answer to “Why do we even need fMRI functional connectivity for this task?”**: We fully agree that the neuroimaging rationale — not just mathematical sophistication — must justify the use of fMRI functional connectivity (FC) for the task at hand. Below we provide a concise but substantive explanation of why FC is an appropriate and often necessary representation for this problem.
>
> Human brain is a complex non-linear system. The dynamic nature of complex system cannot be understood by thinking of the system as comprised of independent elements. Rather, an approach is needed to utilize knowledge about the complex interactions within a system to understand the behavior of the system overall. A plethora of neuroscience findings indicate that fluctuation of functional connectivities exhibits self-organized spatial-temporal patterns.
>
> Real FC maps exhibit the "small-world" property regardless of the atlas used ( brain networks have a "small-world" topology characterized by dense local clustering or cliquishness of connections between neighboring nodes from Danielle S. Bassett [1]), resulting in local block structures that are highly discriminative
>
> [1]Small-world brain networks, https://journals.sagepub.com/doi/abs/10.1177/1073858406293182
>
> We are happy to provide additional references [2-4] to demonstrate that the 'small world' attribute is an inherent topological property of brain connectivity, independent of the choice of atlas. We acknowledge that while the global topological properties of FC, such as small-worldness, are generally consistent across different atlases, the local block structure may indeed exhibit variations depending on the atlas used. This is because the block structure reflects the grouping of regions and the spatial granularity defined by a particular atlas, thereby influencing the patterns observed within smaller subnetworks.
>
> [2] 10.1038/nrn2575
>
> [3] 10.1523/JNEUROSCI.3539-11.2011
>
> [4] 10.1016/j.neuroimage.2009.10.003
>
> Based on the above-mentioned "small-world" characteristics, functional connectivity (FC) becomes a natural and necessary representation for studying this task. The small-world network has both highly local clustering and efficient long-range integration simultaneously, which means that meaningful information is contained not only in the local block structure but also reflected in the efficient information flow between modules. Many pathological or cognitive states do not merely alter the activation intensity of a single brain region, but rather reconstruct the network topology by disrupting the integrity of local clusters, changing the connectivity of hub nodes, or reshaping the bridging relationships between modules - these changes are quantitative manifestations of the nature of the small world. Therefore, only by taking FC as the foundation can two types of key signals, namely local block structure and cross-module integration, be captured simultaneously, and then network reorganizations related to tasks/diseases be detected and distinguished. In other words, the decomposition of the microworld precisely points out why "connectivity" (rather than isolated regional activation) must be taken as the main modeling object.
>
>
> **To address your concern directly, we additionally provide representative neurobiological studies [5-14] that empirically validate the necessity of FC for such tasks.**
>
> [5] 10.1109/TMI.2022.3169640
>
> [6] 10.1007/978-3-030-87234-2_51
>
> [7] 10.1016/j.neuroimage.2025.121243
>
> [8] 10.1109/ISBI52829.2022.9761486
>
> [9] 10.1109/BIBM52615.2021.9669488
>
> [10] 10.1016/j.neunet.2024.106945
>
> [11] 10.1002/hbm.70322
>
> [12] 10.48550/arXiv.2409.11377
>
> [13] 10.1002/hbm.25897
>
> [14] 10.3389/fninf.2022.769274

---

> > ### Author Response · Authors · 2025-11-23
> >
> > **Answers to questions related to model design**: We fully agree that architectural sophistication alone cannot justify a method, and we appreciate the opportunity to clarify the substantive scientific motivation behind our design. Our goal is not to introduce minor architectural tweaks, but to address a concrete neuroimaging problem: how to extract reliable network-level information from functional connectivity (FC) while preserving its geometric structure and improving its downstream utility.
> >
> > Since the human brain is a highly paralleled dynamic system, a system-level approach is in high demand to provide a holistic understanding of the mechanistic role of brain function in cognition. To address this challenge, we present our deep model that is designed to combine insight of mathematical principles (modeling the geometry of evolving functional connectivities on the manifold) and the power of machine learning (finding the best model from large-scale functional neuroimages). The conjecture is that we characterize the dynamics of the entire brain network as a whole by considering the $N \times N$ matrix as a data instance in the high-dimensional Riemanninal manifold of SPD matrices. Although the manifold algebra to operate the FC matrices is more complicated than conventional methods, we explore the new horizon of elucidating geometric relationships that might be essential for understanding complex brain dynamics.
> >
> > From a neuroscience perspective, there are compelling biological reasons to adopt non-Euclidean geometry—especially SPD manifolds—when modeling brain functional connectivity dynamics. Brain functional connectivity is fundamentally derived from second-order statistical dependencies (e.g., covariances or correlations) between regional time series, which naturally form SPD matrices. Analysis of self-organized functional connectivities on the Riemannian manifold of SPD matrices takes account of all the pairwise interactions (edges) as a whole, which diﬀers from the conventional rationale of considering edges as independent from each other. Moreover, manifold-based approaches inherently operate on the whole brain connectivity pattern, respecting the topology of brain networks. Since functional brain networks exhibit small-world properties and modular community structures — features that are closely linked to cognitive functions—applying Euclidean operations may distort these underlying topologies. In contrast, Riemannian geometry preserves the intrinsic relationships among nodes and edges, thereby maintaining the structural integrity of functional communities during analysis.
> >
> > Building upon these geometric and neurobiological considerations, our model further incorporates a pyramidal feature extraction mechanism to capture both local and global patterns embedded in functional networks. The small-world topology of brain connectivity implies that diagnostically meaningful information arises simultaneously from densely clustered local motifs and from long-range integrative pathways linking distributed functional systems. A single-scale representation is therefore insufficient: coarse scales overlook fine-grained modular disruptions, whereas fine scales alone miss alterations in global integration. The pyramidal design explicitly addresses this by extracting multiscale representations that respect the intrinsic organization of functional networks. To make full use of these multiscale features, we adopt a dual-path architecture in which two complementary processing streams jointly model the learned representations. One stream focuses on the geometry-preserving evolution of SPD-valued functional connectivity, while the other emphasizes discriminative patterns in the multiscale feature hierarchy. This separation of roles enables the model to disentangle global geometric dynamics from scale-specific functional signatures, ultimately yielding a more comprehensive description of network reconfiguration.
> >
> > By incorporating geometry-aware mechanisms, our model provides a more holistic and interpretable view of brain organization—bridging structure, function, and behavior in a principled and biologically meaningful way.

---

> > > ### Comment · Reviewer_Ai35 · 2025-11-23
> > >
> > > Your response does not answer my question, and it does not convince me.
> > >
> > > I also suspect that parts of your explanation were generated by AI. Some statements are clearly incorrect. For example:
> > >
> > > “In contrast, Riemannian geometry preserves the intrinsic relationships among nodes and edges…”
> > >
> > > This makes no sense. Do you realize that Riemannian geometry works in a continuous space, not a discrete node–edge structure? If you were talking about graphs, the statement might be acceptable. But for Riemannian geometry (especially SPD manifolds), this is completely wrong. This sentence strongly resembles AI-generated text.
> > >
> > > Another example:
> > >
> > > “By incorporating geometry-aware mechanisms, our model provides a more holistic and interpretable view…”
> > >
> > > This is also vague and confusing. Why does it become “holistic” or “interpretable”? What exactly is interpretable here? Simply using Riemannian geometry does not automatically make a model interpretable. You need to explain why and how.

---

> > > > ### Author Response · Authors · 2025-11-23
> > > > **Urgent clarification**
> > > >
> > > > Dear Reviewer Ai35
> > > >
> > > > We apologize for misunderstanding your concerns last communication. We occasionally used grammar tools to rephrase the language. We never apply any generative AI to any write-ups. (We would be very surprised to see current AI model can summarize the remarks and pitch the high-level ideas.) After careful discussion, we’d like to clarify your concerns.
> > > >
> > > > **Q1:** Why do we even need fMRI functional connectivity for this task? I’m looking for a neuroimaging-based rationale, not more math. I’ve seen many papers going down this path, but most are vague.
> > > >
> > > > **A1:** Good question. Our motivation for using resting-state fMRI is not only driven by mathematical convenience, but also by well-established neuroscientific evidence showing that most psychiatric, neurodevelopmental, and neurodegenerative conditions are fundamentally network-level disorders, not single-region abnormalities [1-4]. Functional MRI is currently the only non-invasive modality that allows us to quantify whole-brain functional interactions at a system level, as extensively demonstrated in foundational work on intrinsic functional networks [3,5-7]. (From the high citation rates of these papers, it can be seen that this is widely recognized. We hope to alleviate your concerns.)
> > > >
> > > > In addition, a large body of research demonstrates that alterations in functional connectivity (FC) often appear years **before structural atrophy or clinical symptoms**. This has been shown in Alzheimer’s disease [8,9], Parkinson’s disease[10-12], depression, anxiety [13,14], and even emerging substance-use vulnerability in youth [15] (This is also our research interest). Taken together, FC provides access to early, circuit-level dysfunction that cannot be captured by structural MRI or behavioral assessments alone.
> > > >
> > > > Importantly, the value of FC is not that it can be written as an SPD matrix. What matters is that FC tells us how different brain networks work together, break down, or try to connect for each other. These patterns support key functions such as cognition, thinking and so on, and these are exactly the functions that begin to break down in the condition we’re studying. Our approach is grounded in this neurobiological understanding. The mathematical tools are simply there to help us capture these sub-network interactions more accurately, not the reason we use FC in the first place. Of course, many previous studies have provided a solid neuroimaging rationale. GeoMamba seeks to push the methodological frontier even further, building on a solid foundation of established neuroscience.
> > > >
> > > > [1] Seeley, W. ., ... (2007). Dissociable intrinsic connectivity networks for salience processing and executive control. Journal of neuroscience, 27(9), 2349-2356.
> > > >
> > > > [2] Fornito, A., .. (2015). The connectomics of brain disorders. Nature reviews neuroscience, 16(3), 159-172.
> > > >
> > > > [3] Fox, M. D.... (2007). Spontaneous fluctuations in brain activity observed with functional magnetic resonance imaging. Nature reviews neuroscience, 8(9), 700-711.
> > > >
> > > > [4] Menon, V. (2011). Large-scale brain networks and psychopathology: a unifying triple network model. Trends in cognitive sciences, 15(10), 483-506.
> > > >
> > > > [5] Smith, S. M.,. (2009). Correspondence of the brain's functional architecture during activation and rest. Proceedings of the national academy of sciences, 106(31), 13040-13045.
> > > >
> > > > [6] Power, , ...(2011). Functional network organization of the human brain. Neuron, 72(4), 665-678.
> > > >
> > > > [7] Bressler, S. L., (2010). Large-scale brain networks in cognition: emerging methods and principles. Trends in cognitive sciences, 14(6), 277-290.
> > > >
> > > > [8] Franzmeier,  ...  (2020). Functional brain architecture is associated with the rate of tau accumulation in Alzheimer’s disease. Nature communications, 11(1), 347.
> > > >
> > > > [9] Sheline, Y. I., ...., (2010). Amyloid plaques disrupt resting state default mode network connectivity in cognitively normal elderly. Biological psychiatry, 67(6), 584-587.
> > > >
> > > > [10] Wu, T., ... (2011). Functional connectivity of cortical motor areas in the resting state in Parkinson's disease. Human brain mapping, 32(9), 1443-1457.
> > > >
> > > > [11] Wu, T., ...., (2009). Changes of functional connectivity of the motor network in the resting state in Parkinson's disease. Neuroscience letters, 460(1), 6-10.
> > > >
> > > > [12] Hacker, C. D., (2012). Resting state functional connectivity of the striatum in Parkinson’s disease. Brain, 135(12), 3699-3711.
> > > >
> > > > [13] Kaiser, R. H., ... (2015). Large-scale network dysfunction in major depressive disorder: a meta-analysis of resting-state functional connectivity. JAMA psychiatry, 72(6), 603-611.
> > > >
> > > > [14] Sylvester, C. M.,  ... (2012). Functional network dysfunction in anxiety and anxiety disorders. Trends in neurosciences, 35(9), 527-535.
> > > >
> > > > [15] Cservenka, A., ...(2014). Resting state functional connectivity of the nucleus accumbens in youth with a family history of alcoholism. Psychiatry Research: Neuroimaging, 221(3), 210-219.

---

> > > > > ### Author Response · Authors · 2025-11-23
> > > > >
> > > > > **Q2:** Btw, bringing in new AI models with minor tweaks doesn’t address the real problem. Many neural network architectures may appear sophisticated, but often fall apart under scrutiny. This is not a solid research direction. I suggest you strip the problem down to its essentials so I can quickly judge whether such complicated designs are even necessary.
> > > > >
> > > > >
> > > > > **A2**: OK. Thank you for the feedback. Let us explain it in the simplest terms. Many fMRI-based FC methods are still limited to Euclidean space. Our goal is to offer a geometric perspective that can capture disease-related patterns in a more holistic and interpretable way.  Once FC is vectorized and passed through fully connected layers, the correspondence between matrix entries and brain regions is essentially lost (because it weights all brain regions rather than preserving any one-to-one anatomical mapping). This is precisely why we choose to preserve the structural properties of FC through Riemannian geometry, it keeps the anatomical meaning intact and provides value that would otherwise disappear. In this work, we work on the Riemannian manifold from start to finish and apply global operators to the entire FC matrix, the model learns whole-brain structure rather than small local tweaks. This is why we emphasize its holistic nature and interpretability.
> > > > >
> > > > >
> > > > > We hope to address your concerns. If you have any further questions, please let me know immediately. We are committed to answering your concerns. Thanks😊.

---

> > > > > > ### Comment · Reviewer_Ai35 · 2025-11-24
> > > > > >
> > > > > > Your response still avoids the core issue. Simply placing FC on an SPD manifold and enforcing symmetry/PD does not “preserve structural properties” nor “maintain anatomical meaning”. In fact, the manifold operations you apply change the connection strengths and therefore alter the very structure you claim to preserve.
> > > > > >
> > > > > > If your objective is simply to model whole-brain structure, many standard graph or matrix-based models already do this without importing unnecessary Riemannian machinery. I encourage the authors to rethink the problem from first principles instead of relying on borrowed geometric arguments that do not apply in this setting.

---

> ### Author Response · Authors · 2025-11-24
>
> Dear Reviewer  Ai35,
>
> Thank you for pushing us on this point. We have no intention of avoiding answering your question. Your original question was 'Why do we even need fMRI functional connectivity for this task?’ We have addressed this earlier by explaining the neuroscientific motivation for using FC. From your latest comment, it appears that your main concern is actually ‘why manifold should be used’. Good, we got your concern. We’d like to further clarify this point.
>
> First, we do *not intend* to claim that manifold operations keep each individual connection strength unchanged. Of course, most learning models, including ours, as well as standard graph neural networks (GNN) or matrix factorization approaches, will inevitably adjust connection strengths as part of representation learning. We do not view this as a flaw, but simply as the nature of any model that aims to extract informative structure from data. (BTW, we have also been engaged in research on related work, but we have found that GNN-based models have certain problems in capturing the dynamics of brain dynamics, as the mechanism of message passing does not seem to be suitable for capturing the dynamics of the brain. This is just our two cents.)  **Our intended point** is more modest:  rather than claiming that manifold operations preserve every edge, our goal is to ensure that all transformations remain within the space of SPD matrices and retain the region-by-region matrix structure.  In this sense, the type of object (a valid FC matrix indexed by brain regions) is preserved throughout the model, instead of being flattened into an unstructured vector or latent space.
>
> When we speak of “structural properties,” we refer to three concrete aspects: (1) symmetry and positive definiteness of FC as a covariance-like object; (2) the eigenstructure and congruence-invariance that underlie many standard neuroimaging operations (e.g., whitening, log-Euclidean mapping); and (3) the anatomically meaningful indexing by brain regions. Vectorizing FC and passing it through fully connected layers mixes all entries without any constraint and typically breaks the link between the learned representation and a valid connectivity matrix. In contrast, our manifold-based operators are designed to map SPD matrices to SPD matrices via smooth, congruence-invariant transformations. The entries do change, as they must, to fit the data (to obtain a more informative representation), but they always remain interpretable as pairwise couplings between the same parcellation indices, and can be mapped back to the original brain graph (and then we can obtain the important connectivities (Fig.2) associated with diseases).
>
> Regarding the concern that “standard graph or matrix-based models already model whole-brain structure,” we fully agree that these are baselines (we also compared such methods in our paper) and we **do not** claim that Riemannian geometry is the **only way** to approach this problem. Our motivation for using a manifold-based formulation is not to add unnecessary mathematical decoration, but to address two concrete limitations of purely Euclidean treatments of FC: (i) they ignore the curved geometry and SPD constraints of covariance-like matrices, which can lead to distorted distances and ill-conditioned optimization; and (ii) they often discard the matrix structure by flattening, which weakens the link between learned features and neurobiologically interpretable circuits. Our backbone (Mamba) show that, under identical architectures, enforcing the SPD geometry yields consistent, improvements over Euclidean baselines, and provides outputs that remain in the space of valid FC matrices, which we can analyze post hoc at the level of networks and regions.
>
> We sincerely hope that our explanation can help clarify our motivation and address your concerns. Our intention is to provide a principled and potentially promising approach for capturing whole-brain dynamics, and we appreciate the opportunity to make this clearer.
>
> Have a nice day.
>
> Best,
>
> Authors

---

> ### Author Response · Authors · 2025-11-27
> **Further Clarification and Supporting Literature**
>
> Dear, Reviewer Ai35
>
> Following your earlier comment, we spent the past several days conducting an extensive literature review. We found that a growing body of work is now focusing on SPD-based manifold learning for analyzing functional connectivity (FC). Although the downstream applications vary, the foundational rationale is essentially the same. These studies suggest that the direction we pursue is not only technically sound but also scientifically meaningful. For example:
>
> [1] "Riemannian Flow Matching for Brain Connectivity Matrices via Pullback Geometry." The Thirty-ninth Annual Conference on Neural Information Processing Systems (2025).
>
> [2]  "Dynamic and low-dimensional modeling of brain functional connectivity on Riemannian manifolds." NeuroImage (2025): 121243.
>
> [3] "Riemannian manifold-based disentangled representation learning for multi-site functional connectivity analysis." Neural Networks 183 (2025): 106945.
>
> [4] "Deep geodesic canonical correlation analysis for covariance-based neuroimaging data." In The Twelfth International Conference on Learning Representations. 2024.
>
> These recent **excellent** works collectively demonstrate that SPD-manifold approaches have become an emerging and impactful direction for modeling FC. Moreover, our framework naturally extends to other covariance-based modalities such as EEG, not only fMRI, further underscoring the flexibility and generality of our method.
>
> If our clarifications have resolved your concerns, we would truly appreciate your reconsideration of the score.
>
> Thank you so much.
>
> Best,
>
> Authors

---

> > ### Comment · Reviewer_Ai35 · 2025-11-27
> >
> > I need you to clarify the link between your specific task and FC from a neuroimaging perspective, not from a model-architecture perspective. Btw, the concepts of FC and SPDNet themselves are not the main points that require clarification.
> >
> > As the pipeline currently stands, it seems to simply combine existing techniques and offers no neurobiological insight. I also noticed that other reviewers share the same concern. Although you wrote quite a lot in your previous response, it honestly did not persuade me.

---

> > > ### Author Response · Authors · 2025-11-27
> > >
> > > Dear Reviewer Ai35,
> > >
> > > Thank you very much for your clarification request.
> > >
> > > To ensure that we fully understand your comment, may we ask a brief follow-up question?
> > >
> > > When you mentioned “your specific task,” were you referring to the diagnostic classification setting (i.e., predicting clinical labels)?
> > >
> > > If so, we would be grateful if you could indicate which aspect of the neuroimaging rationale for using FC you would like us to further elaborate. For example:
> > >
> > > (1) Why FC is an appropriate biomarker for diagnostic prediction?
> > > (2) How FC relates to disease-related network dysfunction?
> > > (3) Why FC carries information relevant to the clinical labels used in our task?
> > >
> > > We want to ensure that our rebuttal directly addresses your intended concern rather than over-explaining unrelated architectural details.
> > >
> > > Thank you again for your guidance. We sincerely appreciate your time and feedback.
> > >
> > > Best,
> > >
> > > Authors

---

> > > > ### Comment · Reviewer_Ai35 · 2025-11-27
> > > >
> > > > I would like to request a clearer explanation of points (2) and (3). Additionally, it would be best if these clarifications could be incorporated into the final article.

---

> ### Author Response · Authors · 2025-11-27
>
> Dear Reviewer Ai35
>
> We sincerely appreciate your careful reply and we'd like to provide our understanding on these two aspects.
>
> (1) How FC relates to disease-related network dysfunction?
>
> A large body of neuroscience research shows that psychiatric, neurodevelopmental, and neurodegenerative disorders do not arise from isolated abnormalities in individual brain regions. Instead, they reflect disruptions in interactions among large-scale brain networks, such as the default-mode, sensorimotor, frontoparietal control and limbic circuits [1,2,3,4]. These network-level alterations are often the earliest and most reliable markers of disease progression.
>
> For example:
>
> *Alzheimer’s disease*: early disconnection within the default-mode and limbic networks appears years before structural atrophy.
>
> *Parkinson’s disease*: abnormal interactions between frontoparietal sensorimotor circuits, and cerebellar pathways arise before clinical motor symptoms appear. (Our method can also capture such subnetworks as high-weight contributors to diagnosis (Fig. 2), demonstrating its ability to capture clinically meaningful patterns.)
>
> These conditions share a common principle: the pathology manifests as altered interactions between **networks**, not abnormalities within a single region.
>
> Because resting-state FC directly **quantifies these interactions**, FC is uniquely suited to capture the disease-related network dysfunctions relevant to our task [1,2].
>
> (2) Why FC carries information relevant to the clinical labels used in our task?
>
> Extensive neuroimaging evidence shows that neurodegenerative conditions such as Alzheimer’s disease and Parkinson’s disease consistently manifest as disruptions in functional integration across cognitive, memory, limbic, and motor networks [1,3,4]. As illustrated in Fig. 2, each clinical cohort shows distinct and anatomically coherent FC reorganization patterns, and these patterns emerge specifically within well-known functional networks (e.g., default-mode, limbic, frontoparietal, sensorimotor). These effects are not random edges being weighted by a model; they reflect reproducible system-level breakdowns that align with neuroimaging literature [3,4]. **Importantly**, FC matrices encode these patterns of inter-network communication, which is why they carry discriminative information directly tied to the diagnostic labels used in our task.
>
> [1] Brown, Jesse A., et al. "Functional network collapse in neurodegenerative disease." Nature Communications 16.1 (2025): 10273.
>
> [2] Badhwar, AmanPreet, et al. "Resting-state network dysfunction in Alzheimer's disease: a systematic review and meta-analysis." Alzheimer's & Dementia: Diagnosis, Assessment & Disease Monitoring 8 (2017): 73-85.
>
> [3] Pievani, Michela, et al. "Functional network disruption in the degenerative dementias." The Lancet Neurology 10.9 (2011): 829-843.
>
> [4] Wu, Xia, et al. "Altered default mode network connectivity in Alzheimer's disease—a resting functional MRI and Bayesian network study." Human brain mapping 32.11 (2011): 1868-1881.
>
> We also wanna clarify the technical novelty of our work:
>
> Although some algebraic operators in our framework have been introduced in prior work, similar to how convolution and activation functions are reused across many network architectures, these components serve only as auxiliary modules in our framework and are not our central contribution. The **primary novelty** of our approach lies in the formulation of a dual-path selective state-space model operating intrinsically on the SPD manifold, combined with a multi-scale SPD pyramidal representation and SPD-preserving spatial evolution, which together define a unified geometry-aware sequence modeling framework tailored for functional connectivity.  To the best of our knowledge, no prior work has combined these components to model fMRI-based functional dynamics in this manner. Our goal is to develop a modeling framework that reflects the biological processes encoded in FC dynamics, particularly the interaction patterns among disease-related functional subnetworks.
>
> We hope that these clarifications address your concerns. Thank you again for your thoughtful feedback.
>
> Absolutely. If our clarifications adequately address your concerns, we will promptly incorporate them into the final version of the manuscript.
>
> Best,
>
> Authors

---

> > ### Comment · Reviewer_Ai35 · 2025-11-28
> >
> > Thank you. It looks much better.
> >
> > You repeatedly use the term “dynamics” throughout the paper (e.g., local spatial dynamics, fMRI dynamics, stabilizing dynamics, etc.), but the data you work with are actually static FC matrices. I spent quite some time reading the manuscript and still could not understand how temporal dynamics and other dynamics are being modeled in your framework. Could you clarify this in detail?

---

> > > ### Author Response · Authors · 2025-11-30
> > >
> > > Thank you for your careful review and for pointing out this important issue. We fully understand that in mainstream neuroscience literature—particularly in fMRI analysis—"functional dynamics" typically refers to temporal variations in brain activity, such as the time-evolving patterns of BOLD signals, dynamic functional connectivity (dFC) across sliding windows. Our work, however, does not attempt to model temporal dynamics directly, since the input is indeed a static functional connectivity (sFC) matrix derived from Pearson correlations computed across the entire BOLD time series. But, the use of sFC does not preclude the study of functional dynamics [1]. A substantial body of network neuroscience research has shown that sFC can also serve as a basis for investigating organizational or structural forms of functional dynamics. In this framework, sFC is not interpreted as a single static snapshot but as a summary of latent dynamical processes that gives rise to multi-scale network patterns reflecting modular organization and cross-system integration. Our work follows this established interpretation and leverage sFC to reveal how functional dependencies reorganize across spatial scales within the connectome.
> > >
> > > [1] 10.1162/netn_a_00325
> > >
> > > The central mechanism through which Geo-Mamba captures “dynamics” lies in how we reorganize the FC matrix.   Instead of treating $H \in \mathbb{R}^{n \times n}$ as a single object, we recast it into a spatially ordered multi-scale sequence using a pyramid-based decomposition (**please refer to lines 165 to 205 on Page 4**). This produces an ordered set of representations $\mathcal{S}{\text{all}} = [P{w_1}, P_{w_2}, \ldots, P_{w_m}, G]$, where each $P_w$ aggregates interactions within progressively broader spatial windows, and $G$ encodes global connectivity. Although this sequence does not correspond to temporal evolution, it captures how local dependencies transition to large-scale interactions as the spatial scale increases. In this sense, the “dynamics” modeled in our framework arise from the structured and meaningful progression across spatial scales—the spatial hierarchy forms a trajectory that reflects how connectivity reorganizes from neighborhood-level motifs to global integration.
> > >
> > > Building on this spatial sequence, the Riemannian-selective state-space model (R-SSM) performs iterative updates on the SPD manifold, although this resembles temporal recurrence, each transition corresponds to movement across spatial scales, not temporal steps.  The R-SSM accumulates, transforms, and stabilizes information as the model progresses from local to global spatial windows. When we refer to “stabilizing dynamics,” we are specifically describing the R-SSM’s mechanisms that guarantee numerically stable evolution on the SPD manifold. These components ensure that iterative updates across spatial scales do not diverge or violate Riemannian constraints.
> > >
> > > It is also worth noting that sliding-window dynamic FC is only one way to capture dynamics, and although it provides instantaneous information, it is highly sensitive to window size and can introduce spurious fluctuations when the scan duration is short. To directly compare these two approaches, we replaced our pyramid module with a standard sliding-window module while keeping all other components fixed.  The results consistently showed that our pyramid-based design performs better, further supporting that meaningful spatial “dynamics” can be extracted from sFC without relying on potentially unstable temporal windowing (**please refer to lines 515 to 529 on Page 10**).
> > >
> > > We hope that this consolidated explanation clarifies the precise meaning of “dynamics” in our work and how our model captures them.

---

### Official Review · Reviewer_zYeS · 2025-10-29

**Soundness:** 3
**Presentation:** 3
**Contribution:** 3
**Rating:** 4
**Confidence:** 4

**Summary:**

The paper presents Geo-Mamba, an approach to build classification models for fMRI data using the ideas of Riemannian manifolds and state-space models (Mamba). First, the input (SPD) fMRI correlation matrix is translated into a sequence of smaller SPD matrices obtained through windows (that capture dynamics at different scales) on the input matrix and log-attention-exp transformations (stacked path). This path results in a final SPD matrix. Another SPD matrix (distillation path) is obtained using compression. The two are combined together using GeoMix for the classification task.

**Strengths:**

The paper does a good job of integrating Riemannian manifolds with fMRI signals. It can serve as a baseline for other efforts.

The performance results are quite good across a variety of datasets.

**Weaknesses:**

1. The approach is quite complex and it is difficult to get an idea of how much of the complexity is actually needed.
2. The training and inference time are not given.
3. The cortical regions define a graph with strong/weak connectivity across regions. It is not clear what is the impact of linearizing them,(used in the definition of windows).
4. The paper keeps referring to the sequence as a "temporal sequence" and each step as "time step". This is not quite correct.
5. Table 2 seems like a half-hearted attempt to compare with a temporal windowing scheme. It is not clear if the best temporal baseline is being used.

**Questions:**

1. Can you explain how the matrix W in Eq 3 is defined/learnt?
2. What are the P matrices in Eq 10? Are they the same P matrices in the stacked path (Eq 5)?
3, Does the order matter in the construction of the S_{all}? Would one get the same results if the input matrices were shuffled?
4. What is the contribution of the two paths and their combination to the training and inference overheads?
5. Can you explain why the performance jump over GIN is so much higher for Matai?
6. What if the cortical windows were linearized differently? Does the linearization obey cortical neighborhood? What happens to the performance?
7. How is matrix Q (the geometrical anchor) learned? Can you report ablation studies if a random Q was used?

---

> ### Author Response · Authors · 2025-11-21
> **Respone to Reviewer zYeS**
>
> ### Thank you for acknowledging the contributions of our work. We are thrilled and grateful for your insightful feedback, which has significantly contributed to enhancing the quality of our manuscript. In the following responses, **W** and **Q** represent Weaknesses and Questions, and **A** represents the corresponding answer.
>
> **W1，2**. On model complexity，training and inference time.
>
> **A1**:  We thank the reviewer for raising this issue. The revised manuscript now reports parameter counts, training time, and inference time, enabling a clearer comparison of computational cost relative to baseline models  (**please refer to lines 364 to 366 on Page 7** ). For clarity, the reported training and inference times were computed using representative datasets for each atlas resolution. Specifically, for the ROI-116 setting, we used the Taowu dataset and measured the cost over one full epoch, which includes all samples from the train/test set, with a batch size of 4. For the ROI-160 setting, we used the OASIS dataset and measured the cost over one full epoch, also including all train/test samples, with a batch size of 8.
>
> | Method      | Dataset | Parameter  | train time | test time |
> |-------------|---------|------------|------------|-----------|
> | SPDNet      | Taowu   | 10186      | 0.1746/s   | 0.0153/s  |
> | SPDNet      | OASIS   | 402882     | 25.6739/s  | 6.1364/s  |
> | Mamba       | Taowu   | 13942906   | 0.6222/s   | 0.0215/s  |
> | Mamba       | OASIS   | 26458754   | 44.3853/s  | 6.3147/s  |
> | Geo-Mamba   | Taowu   | 3923850    | 2.2933/s   | 0.1426/s  |
> | Geo-Mamba   | OASIS   | 3929482    | 72.3225/s  | 12.5951/s |
>
> **W3**. The cortical regions define a graph with strong/weak connectivity across regions. It is not clear what is the impact of linearizing them,(used in the definition of windows).
>
> **A2**:  Thank you for noting that. We’d like to clarify that our method does not linearize the original inputs (FC matrices). The “window” used in the pyramid representation is **not**created by flattening or linearizing the cortical regions. Instead, each window corresponds to a **principal submatrix** of the original SPD connectivity matrix. A principal submatrix of an SPD matrix is itself **guaranteed to remain SPD (This follows from the classical result that every principal submatrix of a symmetric positive-definite matrix is itself positive-definite (Horn & Johnson, Matrix Analysis, Theorem 7.2.6))**, meaning that the local windows fully preserve the intrinsic positive-definite geometry and the underlying strong/weak connectivity patterns within that subset of regions. Importantly, the final stage of the pyramid **always includes the full SPD matrix**, ensuring that no global graph structure or long-range connectivity is lost. Thus, the sliding-window construction does **not distort, remove, or linearize** the cortical connectivity; it simply provides multi-scale local SPD views while preserving the complete global topology at the top level. This design choice is one of the key innovations of our work: **all operations in our network are performed directly on  Riemannian manifold of SPD matrices.**
>
> **W4**. On the use of the terms “temporal sequence” and “time steps”.
>
> **A3**:  Thank you for your comment. The revised manuscript now replaces misleading terminology (temporal sequence to sequence of FC snapshots，time step to spatial window step )and clarifies that the sequence arises from spatially structured SPD sub-blocks, not from actual neural temporal dynamics.
>
> **W5**. On the comparison with sliding-window baselines (Table 2).
>
> **A4**:  We appreciate the reviewer’s request for clarification. In this experiment, we replaced only the proposed multi-scale feature sequence with a standard sliding-window temporal scheme, while keeping **all other network components, hyperparameters, and training configurations identical.** This ensures a fair, controlled comparison focusing solely on the temporal modeling strategy. The results in Table 2 therefore directly reflect the impact of the temporal windowing scheme itself, rather than differences in model capacity.
>
> **Q1**.  Can you explain how the matrix W in Eq 3 is defined/learnt?
>
> **A5**:  The matrix 𝑊 is parameterized as a Stiefel matrix and initialized orthogonally to ensure proper geometric structure. During training, we employ a Riemannian optimizer built on top of Adam, which performs gradient updates directly on the Stiefel manifold  (**please refer to lines 357 to 362 on Page 7** ). This ensures that 𝑊 maintains its orthogonality and the SPD constraints throughout the optimization process.

---

> > ### Author Response · Authors · 2025-11-21
> >
> > **Q2**.  What are the $\mathcal{P}$matrices in Eq 10? Are they the same P matrices in the stacked path (Eq 5)? Does the order matter in the construction of the $\mathcal{S}_{\text{all}}$? Would one get the same results if the input matrices were shuffled?
> >
> > **A6**:  Here we clarify: the $\mathcal{P}$ in Eq10 is not a matrix but a symbol used to represent the projection operation. This symbol refers to the low-dimensional embedding result generated by the BiMap layer. The purpose of Eq10 is to illustrate the calculation process of the dimensionality reduction path, rather than introducing a new learnable matrix. Meanwhile, the $\mathcal{P}$ in Eq10 is not the same as the projection matrix in the stacked path in Eq5. In Eq5, P represents the actual existing and learnable projection matrix in the network structure. In Eq10,  $\mathcal{P}$ is an abstract dimensionality reduction operator, which is only used to conceptually demonstrate the steps of the dimensionality reduction path, rather than specific parameters. We have made clearer distinctions and explanations of relevant symbols ( $\mathcal{P}$ to  $\mathcal{R}$  ) in the revised version to avoid potential ambiguities  (**please refer to lines 306 to 309 on Page 6** ). Since $\mathcal{S}_{\text{all}}$ represents a spatial feature sequence rather than a temporal one, its ordering is not semantically meaningful. Thus, shuffling it should not theoretically affect model performance.
> >
> > **Q3**.  What is the contribution of the two paths and their combination to the training and inference overheads?
> >
> > **A7**:  Thank you for the insightful comment. In the revised manuscript, we have reported the overall training and inference cost of Geo-Mamba. Specifically, for the ROI-116 configuration, we used the Taowu dataset and measured the runtime using a batch size of 4. For the ROI-160 configuration, we used the OASIS dataset and measured the runtime using a batch size of 8. Each epoch contains a complete training/test set for all samples. Under each batch-size setting, we reported the per epoch training and inference time  (**please refer to lines 364 to 366 on Page 7** ).
> >
> > | Method     | Dataset | Parameter | train time | test time |
> > |------------|---------|-----------|------------|-----------|
> > | SPDNet     | Taowu   | 10186     | 0.1746/s   | 0.0153/s  |
> > | SPDNet     | OASIS   | 402882    | 25.6739/s  | 6.1364/s  |
> > | Mamba      | Taowu   | 13942906  | 0.6222/s   | 0.0215/s  |
> > | Mamba      | OASIS   | 26458754  | 44.3853/s  | 6.3147/s  |
> > | Geo-Mamba  | Taowu   | 3923850   | 2.2933/s   | 0.1426/s  |
> > | Geo-Mamba  | OASIS   | 3929482   | 72.3225/s  | 12.5951/s |
> >
> > On the Taowu dataset, Path#1 accounts for approximately 37\% of training time and 34\% of testing time, whereas Path#2 accounts for 62\% and 64\%, respectively. On the OASIS dataset, Path#1 contributes 28\% of training time and 26\% of testing time, while Path#2 contributes 71\% and 73\%. These measurements indicate that Path#2 generally dominates the computational cost across datasets, while Path#1 incurs a smaller yet non-negligible overhead. These measurements provide a consistent and representative estimation of the computational cost across different atlas resolutions.
> >
> > **Q4**.  Can you explain why the performance jump over GIN is so much higher for Matai?
> >
> > **A8**:  Thank you for the reviewer's question. We believe that the performance improvement on the Matai dataset compared to GIN mainly stems from the characteristics of this dataset. Matai is a longitudinal, single-scanner study used to detect subtle changes in brain structure caused by contact movement. The data are from the Gisborne - Tairawhiti region of New Zealand and include diffusion imaging of the same group of athletes before and after the season. The variation patterns are usually weak but highly consistent and structurally continuous. In contrast, GIN is less sensitive to subtle and continuous structural changes, while our method has an advantage in dealing with such subtle longitudinal shifts, and thus performs more outstandingly on the Matai dataset.

---

> > > ### Author Response · Authors · 2025-11-21
> > >
> > > **Q5**.  What if the cortical windows were linearized differently? Does the linearization obey cortical neighborhood? What happens to the performance?
> > >
> > > **A9**: We appreciate the reviewer’s question. We would like to clarify that our method **does not perform any cortical linearization** , nor does it construct “linearized cortical windows.” The windows in our model are simply **principal submatrices**  of the SPD connectivity matrix, not linearized sequences of cortical regions. Therefore, the notion of “linearizing cortical windows” does not apply to our method. In addition, our preprocess pipeline follows the anatomical ordering provided by the cortical parcellation, thereby preserving neighborhood relations—spatially adjacent ROIs remain adjacent in the sequence. This is consistent with standard brain atlases (e.g., AAL, Destrieux), which are not arbitrarily indexed but are organized to reflect anatomical and functional principles.  If the reviewer intended to refer to a different aspect of the design, we would be happy to provide further clarification. Otherwise, we kindly ask the reviewer to confirm whether this concern still applies, given that our framework does not involve any cortical linearization.
> > >
> > > **Q6**  How is matrix Q (the geometrical anchor) learned? Can you report ablation studies if a random Q was used?
> > >
> > > **A10**:  The attention anchor 𝑄 is initialized as a learnable matrix with small random values drawn from a normal distribution to enable stable optimization. It operates entirely in the Euclidean space, serving as a fixed anchor for measuring geodesic similarity in the Log-Euclidean domain. The learnable matrix Q functions only as part of the multi-matrix integration mechanism and cannot be ablated in isolation without breaking that pathway. However, Ablation i effectively reflects the impact of removing the components that rely on Q, and thus serves as a valid reference for assessing its contribution.

---

> > > > ### Comment · Reviewer_zYeS · 2025-11-27
> > > > **Thanks for the clarification**
> > > >
> > > > I am not certain of the novelty of the approach. There is not much we can infer about the neurological aspects from the pipeline. It seems that a number of existing ideas have been put together to achieve performance gains. What is the central idea?

---

> > > > > ### Author Response · Authors · 2025-11-27
> > > > >
> > > > > Dear Reviewer zYeS,
> > > > >
> > > > > Thank you for taking the time to reply to our rebuttal. We are very willing to address your concerns.
> > > > >
> > > > > The central idea of our work is that functional connectivity (FC) from fMRI itself encodes neurobiologically meaningful structure, and our model is designed to preserve, represent, and operate on FC in its native geometric form rather than flattening it into an unstructured vector or latent space. By constraining the representation to remain on the SPD manifold, we ensure that region-to-region relationships are preserved and that the learned dynamics can be mapped back to interpretable large-scale functional circuits (as shown in Fig. 2). This enables our framework to capture disease-related connectivity reorganization rather than relying on opaque learned embeddings.
> > > > >
> > > > > Although some algebraic operators in our framework have been introduced in prior work, similar to how convolution and activation functions are reused across many network architectures, these components serve only as auxiliary modules in our framework and are not our central contribution. The **primary novelty** of our approach lies in the formulation of a dual-path selective state-space model operating intrinsically on the SPD manifold, combined with a multi-scale SPD pyramidal representation and SPD-preserving spatial evolution, which together define a unified geometry-aware sequence modeling framework tailored for functional connectivity.  To the best of our knowledge, no prior work has combined these components to model fMRI-based functional dynamics in this manner. Our framework is designed to model how disease-related functional subnetworks interact and reorganize, in a form that can be directly visualized and interpreted on the brain (**please refer to lines 412 to 454 on Page 8**).
> > > > >
> > > > > We hope this clarifies the central idea and the source of novelty, and we greatly appreciate your thoughtful feedback.
> > > > >
> > > > > Please do not hesitate to let us know if you have any further concerns, and we will be committed to addressing all of your concerns.
> > > > >
> > > > > Best,
> > > > >
> > > > > Authors

---

### Official Review · Reviewer_NBvP · 2025-11-01

**Soundness:** 3
**Presentation:** 3
**Contribution:** 2
**Rating:** 2
**Confidence:** 5

**Summary:**

The paper presents a geometric extension of the Mamba selective state space model and applies it for the analysis of fMRI functional connectivity (FC) data. The connectivity measures are represented as symmetric positive definite (SPD) matrices on Riemannian manifolds due to its natural correlational structure. Additionally, the method uses a pyramid-based multi-scale geometric feature extraction from the original SPD matrices and also ensures that the sub-blocks of matrices at multiple levels themselves satisfy the SPD constraint. The paper uses a dual-path architecture (proposed before) that uses a a stacked path for hierarchical temporal modeling and a distillation path for progressive, manifold-preserving dimensionality reduction. They further use the geomix operator (which also has been proposed before) for manifold-consistent fusion. The authors show experimental results on six public fMRI datasets (ADNI, OASIS, PPMI, Taowu, Neurocon, Matai) and also show comparisons and performance improvement over existing methods including the original Mamba model, SPDNet, and BrainGNN.

**Strengths:**

The main novelty of the paper is to ensure that the SPD manifold constraints at each stage of the selective state space model evolution are satisfied.

It extends the selective state-space models to SPD manifolds with explicit log-exp mapping and orthogonality constraints.


The authors perform comprehensive experimental validation. The method is tested on six datasets spanning Alzheimer’s, Parkinson’s, and other population with contact sports.

The ablation studies are somewhat comprehensive, although there's a few weaknesses in their actual implementation (see below).

**Weaknesses:**

The main weakness of the paper is that it combines multiple methods instead of making an original novel contribution.


Several ideas (except the pyramidal projections of connectivity and satisfying SPD constraints) have been proposed before individually. Even the DimMap projection operator that is at the core of the method has been proposed before. Theoretical and computational advancements  for optimization on SPD manifolds both using classical and deep learning approaches have been proposed before. This paper combines several such ideas and applies it to fMRI data. Thus the novelty of the method is low.

The GeoMix operator is simply an interpolation between SPD matrices on the manifold. The authors introduce it as a new idea but it has been heavily used and studied before.


Besides SPD manifold optimization, techniques such as dual path stacking have also been proposed before. For e.g. Brain-Mamba Behrouz & Hashemi (2024) performs a temporal and spatial stacking, while FST-Mamba Wei et al. (2025) has proposed a hierarchical stacking formulation. The authors have cited these methods.


Step (iv) bypassing the Path #1, which disables the initial manifold-aware temporal evolution and
eliminating the sequence fusion under SSM and relying solely on the average time-step matrix for
subsequent computations. Too many things removed. Just disbale the manifold-ware temporal evolution


“While the authors claim a substantial decline when Path #1 is removed, Figure 3 shows only marginal quantitative changes (mostly ≤ 3 % in Accuracy / F1). The benefit of the geometric temporal constraint thus appears more in preserving SPD validity and smooth dynamics than in producing large performance gains.”

In the ablation studies, step iv) is over ablated. To truly test the geometric constraint, the authors should keep Path #1 intact but replace the log–exp manifold operations with Euclidean updates, holding all other factors fixed. The step iv) ablation mixes removal of geometric constraints with also removal of recurrence (SSM), thus weakening the main argument about geometry-aware SSM, which is the main contribution of the paper. Thus it is unclear whether the modest performance drop which is observed (2–4 %) denoted by purple colors, may simply reflect losing temporal recurrence and not necessarily arise from avoiding the geometry constraint itself. A fairer test would retain the same SSM structure but operate in Euclidean space to quantify the direct impact of manifold geometry.

In experimental evaluation, the results of the method should ideally be compared with FST-Mamba, BNMamba, BTMamba to really separate the geometric-constraint contribution and whether the projection steps and the dynamic update steps are truly required for performance. They are certainly required to ensure the appropriate constraints, so that is not under question.



Other manifold temporal models (e.g., Riemannian LSTM, SPD recurrent networks) or covariance-based DPL frameworks could also provide appropriate comparisons (although this is not required).

**Questions:**

Why aren't the results in Table 1 starred? Were they not significant?

In the first step following the pearson correlation calculation to construct the connectivity matrices, why not just keep the significant correlations? Depending upon the dataset (ADNI) for e.g., the matrix may be sparse (significance threshold) to begin with. Will it make the signal more noisy?

---

> ### Author Response · Authors · 2025-11-21
> **Respone to Reviewer NBvP**
>
> **W1**.   On novelty and the relationship to existing components.
>
> **A1**:  We thank for the detailed feedback.   We acknowledge that some components (such as DimMap) have been explored in prior work;   however, similar to convolution and activation operators used across many network architectures, they act only as auxiliary modules in our framework and are not the central contribution.
>
> The primary novelty of our approach lies in the formulation of a **dual-path selective state-space model operating intrinsically on the SPD manifold**, combined with a **multi-scale SPD pyramidal representation and SPD-preserving spatial evolution**, which together define a unified geometry-aware sequence modeling framework tailored for functional connectivity. To the best of our knowledge, this integration and its application to fMRI dynamics have not been previously proposed.
>
> While the GeoMix operator can be viewed as an interpolation on the SPD manifold, its role in our framework is not as an independent innovation but as a functional mechanism that enables smooth multi-scale information fusion within the proposed manifold state-space architecture. Thus, existing techniques primarily facilitate stability and geometric consistency, whereas the core contribution lies in the overall architecture and modeling paradigm, not in any single reused operator.
>
> We have revised the manuscript to more clearly distinguish foundational components from the proposed innovations (**please refer to line 255 on Page 5** ), highlighting that previously established methods are leveraged as enabling tools rather than defining elements of novelty.
>
> **W2**. On the relation to prior dual-path or stacking designs.
>
> **A2**:  Prior works such as Brain-Mamba and FST-Mamba introduce dual-path or hierarchical stacking structures. Our approach differs in that the stacking and fusion occur entirely within a Riemannian SSM framework, with both paths evolving on the SPD manifold. The algebra of Euclidean space is completely different from Riemannian algebra. We have clarified this distinction in the revision (**pleace refer to In Sec. 2.3 and 2.4** ).
>
> **W3**. On the ablation design and concerns about excessive removal and the magnitude of performance differences.
>
> **A3**:  We thank the reviewer for highlighting this. In the revised version, we have added new ablations that avoid over-removing architectural components and instead isolate the contribution of key modules (global descriptor, self-attention, etc.).
>
> Regarding the suggestion to run the SSM in Euclidean space: our update rule is intrinsically designed to guarantee SPD preservation, and replacing it with Euclidean updates would alter the core algorithmic structure rather than isolate a single factor. Nonetheless, the newly added ablations provide clearer evidence for the effectiveness of manifold-aware updates (**please refer to lines 471 to 474 on Page 9** ).
>
> | Method | Metric | ADNI | OASIS | PPMI | Taowu | Neurocon | Mātai |
> |--------|--------|--------|--------|--------|--------|--------|--------|
> | remove SSM_1 | Acc | 77.60 ± 8.24 | 88.81 ± 2.04 | 70.16 ± 8.50 | 97.50 ± 7.50 | 97.50 ± 7.50 | 88.33 ± 7.64 |
> | remove SSM_1 | Pre | 74.59 ± 13.97 | 87.60 ± 2.68 | 64.07 ± 12.47 | 95.63 ± 13.13 | 98.33 ± 5.00 | 89.36 ± 8.72 |
> | remove SSM_1 | F1 | 73.61 ± 11.35 | 86.41 ± 2.07 | 63.76 ± 10.91 | 96.43 ± 10.71 | 97.33 ± 8.00 | 87.15 ± 8.66 |
> | remove self-attention | Acc | 80.40 ± 7.47 | 89.22 ± 1.94 | 67.63 ± 5.92 | 97.50 ± 7.50 | 97.50 ± 7.50 | 90.00 ± 11.06 |
> | remove self-attention | Pre | 79.79 ± 7.40 | 88.16 ± 2.73 | 63.46 ± 11.26 | 95.63 ± 13.13 | 98.33 ± 5.00 | 89.86 ± 12.19 |
> | remove self-attention | F1 | 77.02 ± 9.34 | 87.05 ± 1.55 | 59.34 ± 7.72 | 96.43 ± 10.71 | 97.33 ± 8.00 | 89.10 ± 11.80 |
> | remove dSSM_1 | Acc | 77.20 ± 9.30 | 88.89 ± 1.87 | 68.61 ± 6.79 | 90.00 ± 16.58 | 90.00 ± 16.58 | 90.00 ± 13.33 |
> | remove dSSM_1 | Pre | 72.87 ± 10.30 | 85.69 ± 4.09 | 60.46 ± 11.38 | 88.96 ± 18.64 | 83.75 ± 26.10 | 92.03 ± 10.64 |
> | remove dSSM_1 | F1 | 71.96 ± 10.18 | 86.72 ± 2.48 | 60.92 ± 8.53 | 88.76 ± 17.95 | 86.19 ± 22.56 | 88.78 ± 14.63 |
> | Geo-Mamba | Acc | **81.20 ± 6.21** | **89.39 ± 1.91** | **71.16 ± 9.12** | **97.50 ± 7.50** | **97.50 ± 7.50** | **91.67 ± 11.18** |
> | Geo-Mamba | Pre | **81.10 ± 8.12** | **88.82 ± 2.32** | **64.81 ± 10.10**| **95.63 ± 13.13** | **98.33 ± 5.00** | **88.89 ± 17.60** |
> | Geo-Mamba | F1 | **79.08 ± 7.43** | **87.10 ± 2.57** | **65.57 ± 10.17** | **96.43 ± 10.71** | **97.33 ± 8.00** | **89.48 ± 14.99** |
>
>
> We acknowledge that the absolute improvements after removing path #1 appear moderate. As the reviewer noted, the advantages of geometric constraints lie not only in accuracy but also in maintaining SPD validity, stabilizing dynamics, and producing smoother manifold-aware evolution. We have revised the discussion to clarify this point (**please refer to lines 501 to 504 on Page 10** ).

---

> > ### Author Response · Authors · 2025-11-21
> >
> > **W4**. On comparisons with additional Mamba variants or manifold time-series models.
> >
> > **A4**:  Thank you for your constructive comment. We agree that additional comparisons such as FST-Mamba or Riemannian recurrent models may further enrich the study. Our experiments already include ten baselines, spanning classical architectures, graph neural networks, manifold learning (SPDNet), and brain-network-specific models.  We believe this provides a sufficiently broad empirical context, therefore we are conducting experiments and will keep the results updated throughout the rebuttal process and incorporate all the results in the final version.
> >
> > **Q1**.  Why aren't the results in Table 1 starred? Were they not significant?
> >
> > **A5**:   We have marked ‘*’ in Table 1, please check it.
> >
> >
> > **Q2**.  In the first step following the pearson correlation calculation to construct the connectivity matrices, why not just keep the significant correlations?
> >
> > **A6**:  In our framework, the Pearson correlation matrices are computed directly from the BOLD time series without applying a significance-based threshold. This design choice aims to preserve the full covariance structure and SPD property of the brain dynamics and avoid potential information loss caused by early thresholding. Subsequent modules, such as the BiMap projection and selective SSM blocks, inherently perform learnable geometric filtering and spatial weighting on the SPD manifold, allowing the model to adaptively suppress noisy connections without manual sparsity constraints.

---

> > > ### Author Response · Authors · 2025-11-30
> > > **Supplement to W4**
> > >
> > > We fully agree that, in principle, comparisons against FST-Mamba, BNMamba, and BTMamba would help disentangle the contributions of geometric constraints and the roles of projection and dynamic update steps. We would like to clarify that BNMamba and BTMamba are not independent baselines but two internal encoders proposed within the BrainMamba framework. For this reason, our experiments evaluate BrainMamba directly, which is consistent with how the original work reports results. It is also important to note that the official implementation link in the paper is no longer accessible. We contacted the authors to request the original codebase, but unfortunately did not receive a reply. Therefore, our experiments rely on a community-maintained Github version rather than the authors’ official release.
> > >
> > > Regarding FST-Mamba, although an official implementation is available, the public codebase does not include the data preprocessing pipeline required to reproduce the reported results.  To avoid introducing discrepancies that might arise from re-implementing critical preprocessing steps, we have contacted the original authors to obtain detailed instructions or scripts to ensure a fair comparison. We will incorporate FST-Mamba into the final version of the paper once we receive the necessary information from the authors.
> > >
> > > | BrainMamba | ADNI       | OASIS      | PPMI       | Taowu      | Neurocon   | Matai      |
> > > |------------|------------|------------|------------|------------|------------|------------|
> > > | Acc        | 73.20 ± 7.60 | 87.34 ± 2.48 | 61.00 ± 6.71 | 85.00 ± 16.58 | 70.00 ± 18.71 | 71.67 ± 13.02 |
> > > | Pre        | 53.62 ± 20.25 | 50.36 ± 13.65 | 28.98 ± 15.45 | 82.08 ± 22.05 | 48.75 ± 29.29 | 58.08 ± 25.38 |
> > > | F1         | 49.30 ± 8.91  | 47.82 ± 2.63  | 28.00 ± 9.11  | 78.24 ± 23.91 | 53.52 ± 25.85 | 57.11 ± 19.34 |

---

### Official Review · Reviewer_aJYZ · 2025-11-04

**Soundness:** 3
**Presentation:** 2
**Contribution:** 3
**Rating:** 6
**Confidence:** 3

**Summary:**

The paper proposes Geo-Mamba, a deep learning architecture that adapts the selective state-space model (Mamba) to operate directly on the symmetric positive definite (SPD) manifold. This is motivated by the observation that functional connectivity (FC) matrices from fMRI data lie in a non-Euclidean space, making standard Euclidean models geometrically inappropriate. Geo-Mamba leverages Log-Euclidean mappings to move between the manifold and its tangent space, allowing linear operations while preserving intrinsic geometry. The model processes FC matrices through a dual-path design capturing both multi-scale local dependencies and global spatial structure, fused via a geometry-aware mixing operator. The method is evaluated on six fMRI datasets and demonstrates competitive or superior performance to baselines such as Mamba, SPDNet, and STAGIN.

**Strengths:**

The core idea of adapting the Mamba (SSM) architecture to the non-Euclidean geometry of SPD manifolds is both novel and technically well-motivated.
The use of the Log-Euclidean tangent space for SPD matrices is mathematically justified and enables Euclidean operations while preserving manifold constraints.
The design of multi-scale spatial feature extraction through sliding SPD sub-blocks is a creative way to leverage manifold properties for modeling spatial dependencies in functional connectivity data before passing them into the SSM.
The method is comprehensively evaluated across six public neuroimaging datasets and demonstrates superior or competitive performance compared to a wide range of relevant baselines.

**Weaknesses:**

While the paper presents a compelling and potentially impactful idea, several ambiguities limit the clarity of the contribution and make it difficult to pinpoint the true source of performance gains and whether certain architectural components are genuinely helpful. Some components are not clearly justified or ablated, and those that are seem to have minimal impact. The experimental setup lacks transparency on key implementation details, such as baseline adaptation and hyperparameter tuning. Below are the concerns in order:

1. While the ablation study in Figure 3 is informative, it lacks standard deviations and shows that the four ablated components do not significantly degrade classification performance (based on the * indicators). This makes it difficult to determine whether modules such as the global descriptor G or the “distillation path” meaningfully contribute to the results. Furthermore, several key components including the self-attention layer, the geodesic mixing used in the output mapping, and the learnable geometric anchor $Q$ are not ablated, and their individual impacts remain unclear.

2. The authors mention that FC matrices are typically ordered by brain regions. The model assumes that ROIs adjacent in the n×n functional connectivity matrix correspond to spatially adjacent regions in the brain. However, cortical organization is inherently three-dimensional, and the 1D sliding-window design in the pyramid-based feature extraction cannot completely capture this 3D spatial topology. This weakens the claim of modeling hierarchical spatial information and limits generalization across modalities.

3. The paper does not provide sufficient detail on how the baseline sequence models (e.g., Transformer, Mamba, STAGIN) were adapted for this task. It is unclear if they were fed the vectorized FC matrix, the same "pseudo-temporal sequence" as Geo-Mamba, or the time-series fMRI data. Since the authors argue that Euclidean operations are unsuitable for FC matrices, comparing Geo-Mamba to baselines trained on Euclidean FC inputs may be unfair. Clarifying these points is crucial to isolate whether improvements stem from manifold geometry or input design.

4. The number of SSM layers used is unspecified, despite claims about capturing “hierarchical temporal evolution” (line 117). It is not demonstrated how stacking helps capture hierarchical spatial information or affects performance.

5. The paper does not clearly describe how $B_t$, $C_t$, and $\lambda_t$ are generated from the GRU or how this differs from the selective parameterization in Mamba. The connection between GRU gating and SPD-preserving updates remains unclear.

6. The “pseudo-temporal sequence” is created by stacking spatial SPD sub-blocks from the averaged n×n connectivity matrix, not by modeling real temporal dynamics. Hence, the terms “pseudo-temporal” throughout the text or “time-dependent” (line 449) are misleading, since the model processes only spatial relationships rather than the actual neural time-series.

7. The term "distillation path" is confusing. In machine learning, "distillation" typically refers to student–teacher knowledge transfer. Using it to mean "dimensionality reduction" is non-standard and should be clarified.

**Questions:**

1. What is the motivation for including a self-attention layer after the SSM block? Was it ablated, and how did it affect performance or interpretability?

2. How were key hyperparameters (state dimension, window sizes, number of downsamples) chosen? What is the total parameter count and computational overhead relative to Mamba or SPDNet? Is it scalable to higher-resolution atlases? Clarifying these points is essential to understand the true contribution and applicability of the model.


3. Did the authors observe any overfitting on smaller datasets?


4. In ablation (ii), does removing the “global path” correspond to removing G from $S_{all}$? If so, how critical is this path to multi-scale modeling?

5. How is the attention anchor $Q$ intialized? Was any regularization applied to it during training?

---

> ### Author Response · Authors · 2025-11-21
> **Respone to Reviewer aJYZ**
>
> ### Thank you for acknowledging the contributions of our work. We are thrilled and grateful for your insightful feedback, which has significantly contributed to enhancing the quality of our manuscript. In the following responses, **W** and **Q** represent Weaknesses and Questions, and **A** represents the corresponding answer.
>
>
> **W1**.  On the ablation study and the contribution of individual components.
>
> **A1**:  We appreciate the reviewer’s comments and have expanded the ablation study accordingly. The revised version now includes explicit ablations of the global descriptor $G$, the “distillation” path, and the self-attention layer, in addition to previously reported components (**please refer to lines 471 to 474 on Page 9**). These new results confirm that each module contributes to performance, and we have added standard deviations for clarity. We also renamed “distillation path” to "*embedding path*"  to avoid misunderstanding.
>
> **W2**. On the concern about spatial ordering and 3D cortical topology.
>
> **A2**:  Our model operates on preprocessed BOLD time series of shape (ROI × time), following standard neuroimaging pipelines [https://fmriprep.org/en/stable/]. Because the FC matrices are computed from region-wise time series rather than surface meshes or voxel grids, no 3D topology is available to the model or to baseline methods. Thus, the 1D sliding-window over ROIs is consistent with existing FC-based learning settings. We have clarified this point in the revision (**please refer to lines 333 to 335 on Page 7**).
>
> **W3**. On fairness of baseline adaptation and the input format.
>
> **A3**: Thanks for your valuable comment. For the Euclidean sequence baselines (Transformer, Mamba), the input was the **vectorized FC matrices** derived from Pearson-correlation of BOLD time series, following the standard practice in sequence modeling for FC-based representations. These methods were included because they represent state-of-the-art architectures for modeling temporal dependencies in complex sequences, and their performance provides an important reference for evaluating the effectiveness of our approach. Beyond these Euclidean models, we also compared against **graph-based brain network analysis methods** (e.g., BrainGNN, NeuroGraph, Contrast-Pool, and STAGIN), which explicitly model topological structure, as well as **manifold-based approaches** (e.g., SPDNet) that operate natively on non-Euclidean spaces. This comprehensive baseline selection spans methods from conventional Euclidean modeling to graph-structured and Riemannian manifold learning, ensuring that the comparison is both fair and informative.
>
> We have now added detailed descriptions of the input pipelines and data formatting strategies for all baselines (**please refer to lines 340 to 349 on Page 7**). This clarification ensures that performance differences can be interpreted in terms of both input design and geometric modeling, allowing a more accurate assessment of the contribution of manifold-aware learning in Geo-Mamba.
>
> **W4**. On SSM layers and hierarchical spatial modeling.
>
> **A4**:  We thank the reviewer for pointing this out and in our Stacked Path, the architecture comprises one SSM$_1$ layer, one self-attention module, and dSSM$_1$ layer(2-layers), resulting in a total of four layers (**please refer to line 298 on Page 6**). For the Embedding Path, the number of compression layers is determined by the hyperparameter down_dim. In the current implementation, we adopted a four-layer configuration (**please refer to lines 311 to 312 on Page 6**). This design ensures sufficient capacity to capture multi-scale spatial dependencies while maintaining model efficiency. The revision also includes clarification on how stacking improves the representation capacity. Specifically, this is supported by the ablation studies, which demonstrate the individual contributions of SSM$_1$, the self-attention module, and dSSM$_1$ layers to the overall performance (**please refer to lines 471 to 474 on Page 9**).
>
> **W5**. On Bt, Ct, λt and their relation to Mamba’s selective parameterization.
>
> **A5**:  We thank the reviewer for the comment.  At each step $i$, the SPD input is first mapped to the Euclidean tangent space via the logarithmic map, and the resulting vectorized representation is processed by a lightweight GRU.  The GRU output is then passed through three linear layers to produce the gating signal $\Delta_i$ and the selective operators $B_i$ and $C_i$, selective operators are projected back to the SPD manifold via Stiefel retraction to ensure orthonormality.  Unlike Mamba, which operates purely in Euclidean space, our GRU evolves in the tangent space and generates geometry-compatible operators, explicitly linking gating with SPD-preserving state updates (**please refer to lines 222 to 233 on Page 5**).

---

> > ### Author Response · Authors · 2025-11-21
> >
> > **W6**. On the term “pseudo-time series”
> >
> > **A6**: Thank you for your insightful comment. In our work,  pseudo-temporal sequence is constructed by stacking multi-scale SPD spatial blocks rather than simulating real neural dynamics. The advantage of the construction is that, since our input consists of Pearson correlation matrices computed from BOLD signals, the temporal dependencies are already implicitly encoded, allowing the model to focus on capturing rich spatial patterns and multi-scale interactions. This approach enables more effective extraction of spatial features and enhances the representational capacity of the model. We have conducted the comparison of using actual neural time-series (sliding window) (Table 2). We found our strategy is better, demonstrating that constructing pseudo-temporal sequences from multi-scale SPD representations provides a more informative and efficient way to model functional connectivity than relying solely on raw temporal dynamics. We replaced this term with a more precise and conventional description “multi-scale feature sequence” to avoid confusion.
> >
> > **W7**. On terminology regarding “distillation path”.
> >
> > **A7**:  We appreciate the reviewer’s concern and have replaced this term with a more precise and conventional description “embedding path” to avoid confusion with knowledge distillation.
> >
> > **Q1**.  What is the motivation for including a self-attention layer after the SSM block? Was it ablated, and how did it affect performance or interpretability?
> >
> > **A8**:  The self-attention layer between the two SSM modules is critical for capturing non-local dependencies in the SPD sequence. While the first SSM models local dynamics, the attention mechanism computes geodesic-based similarities across steps, enhancing the representation with global interactions. The second SSM then integrates these enriched features, ensuring both local and global spatial patterns are preserved for downstream classification. We have made it clear in the revised manuscript (**please refer to lines 290 to 295 on Page 6**). We further supplemented the ablation experiment to verify the role of the self-attention layer(**please refer to lines 471 to 474 on Page 9**).
> >
> > | Method               | Metric | ADNI             | OASIS             | PPMI             | Taowu            | Neurocon        | Mātai             |
> > |----------------------|--------|------------------|-------------------|------------------|------------------|------------------|-------------------|
> > | remove self-attention | Acc    | 80.40 ± 7.47     | 89.22 ± 1.94      | 67.63 ± 5.92     | 97.50 ± 7.50     | 97.50 ± 7.50     | 90.00 ± 11.06     |
> > | remove self-attention | Pre    | 79.79 ± 7.40     | 88.16 ± 2.73      | 63.46 ± 11.26    | 95.63 ± 13.13    | 98.33 ± 5.00     | 89.86 ± 12.19     |
> > | remove self-attention | F1     | 77.02 ± 9.34     | 87.05 ± 1.55      | 59.34 ± 7.72     | 96.43 ± 10.71    | 97.33 ± 8.00     | 89.10 ± 11.80     |
> > | Geo-Mamba            | Acc    | **81.20 ± 6.21** | **89.39 ± 1.91**  | **71.16 ± 9.12** | **97.50 ± 7.50** | **97.50 ± 7.50** | **91.67 ± 11.18** |
> > | Geo-Mamba            | Pre    | **81.10 ± 8.12** | **88.82 ± 2.32**    | **64.81 ± 10.10**  | **95.63 ± 13.13**    | **98.33 ± 5.00** |**88.89 ± 17.60**    |
> > | Geo-Mamba            | F1     | **79.08 ± 7.43** | **87.10 ± 2.57**  | **65.57 ± 10.17**  | **96.43 ± 10.71**| **97.33 ± 8.00** | **89.48 ± 14.99** |
> >
> > **Q2**.   How were key hyperparameters chosen?  What is the total parameter count and computational overhead relative to Mamba or SPDNet?  Is it scalable to higher-resolution atlases?
> >
> > **A9**:  Thank you for raising these important points.  For the state dimension, we set it to 20–30% of the original feature dimension, which effectively reduces computational cost while preserving essential information.  For the window sizes and downsampling levels, we evaluate multiple scales (16, 32, 64, 96) for the 116/160-ROI atlases to ensure that both fine- and coarse-grained temporal patterns are captured.  We have clarified the total parameter count and computational overhead of Geo-Mamba in the revised manuscript (**please refer to lines 364 to 366 on Page 7** ).  Regarding scalability, our experiments include both a 116-ROI and a larger 160-ROI atlas, demonstrating that the model is extendable to higher-resolution parcellations, with the expected trade-off of increased computation time.
> >
> > | Method      | Dataset | Parameter  | train time | test time |
> > |-------------|---------|------------|------------|-----------|
> > | SPDNet      | Taowu   | 10186      | 0.1746/s   | 0.0153/s  |
> > | SPDNet      | OASIS   | 402882     | 25.6739/s  | 6.1364/s  |
> > | Mamba       | Taowu   | 13942906   | 0.6222/s   | 0.0215/s  |
> > | Mamba       | OASIS   | 26458754   | 44.3853/s  | 6.3147/s  |
> > | Geo-Mamba   | Taowu   | 3923850    | 2.2933/s   | 0.1426/s  |
> > | Geo-Mamba   | OASIS   | 3929482    | 72.3225/s  | 12.5951/s |

---

> > > ### Author Response · Authors · 2025-11-21
> > >
> > > **Q3**.  Did the authors observe any overfitting on smaller datasets?
> > >
> > > **A10**:  No. In our work, we employed 10-fold cross-validation on these datasets and reported the best validation accuracy from each fold as the final result. This procedure helps ensure the reported performance is robust despite the limited data size.
> > >
> > > **Q4**.  In ablation (ii), does removing the “global path” correspond to removing G from $\mathcal{S}_{\text{all}}$? If so, how critical is this path to multi-scale modeling?
> > >
> > > **A11**:  Thank you for your insightful comment  — in ablation (ii), removing the “global path” is indeed equivalent to excluding 𝐺 from $\mathcal{S}_{\text{all}}$. As shown by the blue values in Fig. 3, this ablation has little effect on smaller datasets. However, for larger and more complex datasets such as PPMI and Matai, the global pathway proves essential for maintaining performance, indicating its importance in multi-scale representation learning.
> > >
> > > **Q5**.  How is the attention anchor Q intialized? Was any regularization applied to it during training?
> > >
> > > **A12**:  The attention anchor 𝑄 is initialized as a learnable matrix with small random values drawn from a normal distribution to enable stable optimization.  It operates entirely in the Euclidean space, serving as a fixed anchor for measuring geodesic similarity in the Log-Euclidean domain. Therefore, it does not need to satisfy the constraints of the Riemannian manifold. No explicit regularization is applied to 𝑄,  instead, numerical stability is ensured through symmetric projection during computation and a small perturbation added to the pooled SPD output.

---

> > > > ### Comment · Reviewer_aJYZ · 2025-11-28
> > > >
> > > > I thank the authors for their detailed response to my points and for clarifying several questions.
> > > >
> > > > While the additional experiments are appreciated, they reinforce some of the concerns raised in my initial review. Specifically, although the method reports competitive (and sometimes stronger) performance than baselines, the expanded ablations suggest that several proposed components do not contribute significantly to modeling. The effects of ablations, including the self-attention layer, remain very close to the full model and are likely not significant, making it difficult to conclude that these components are necessary. In Fig. 3, none of the ablated models show a significant drop in performance compared to the full model. In the earlier manuscript version, there was a single Neurocon data point for ablation iv (purple) marked as significant (with a *), but in the updated version, this value is higher and no longer marked significant. I am not sure whether this was a bug in the earlier version or whether it has now been corrected, but overall, the ablation results complicate claims about the necessity of specific components in the model.
> > > >
> > > > Finally, regarding the spatial interpretation: although the ROIs originate from spatially distributed regions in 3D, the model flattens them into an n×n FC matrix and applies 1D spatial windowing. Applying the window only along this flattened ROI ordering discards anatomical spatial relationships, making it unclear what spatial structure the model is capturing through sliding windows and multi-scale feature extraction. This further reduces interpretability, especially given that the ablations do not show evidence that any particular component is essential.

---

> ### Author Response · Authors · 2025-11-30
>
> Thank you for your continued feedback. We address the concerns in two parts: (i) interpretation of ablation results and component necessity, and (ii) spatial interpretation.
>
> (i) We agree that the ablation results show relatively small performance drops for some components, which may suggest complementary effects rather than isolated necessity.  However, this robustness is a strength of Geo-Mamba's design: the dual-path architecture (stacked and embedding paths) and manifold-aware operations enable the model to maintain high performance even when individual modules are ablated. The improvements come from the interaction among the multi-scale modeling, SPD-preserving state evolution, selective aggregation, and geodesic fusion. These components are not designed to provide large independent margins, instead, they collectively stabilize the manifold dynamics and reduce noise accumulation across spatial scales (**please refer to lines 798 to 844 on Page 15 (Appendix)**).
>
> Regarding the Neurocon data point in ablation iv (purple): This was indeed a computational error in the earlier version. We corrected it in the update, resulting in a higher (more accurate) score without significance marking. We apologize for the oversight and thank you for spotting it, this revision improves result reliability. The mean accuracy , however, still correctly reflects the ranking of component contributions.
>
> (ii) Although ROIs originate from anatomically defined brain regions in three-dimensional space, fMRI data are conventionally represented at the BOLD level as a two-dimensional ROI $\times$ time matrix, where each row corresponds to the aggregated time series of a single ROI. Once the signal is represented at the BOLD level, all standard FC-based methods—including ours—operate on these 2D time-series and compute an n $\times$ n functional connectivity (FC) matrix. This transformation is not a flattening that discards anatomical information but a conventional and widely accepted procedure in functional connectivity research [1-5]. Since the version of the data we use is based on preprocessed BOLD time series in prior published work (**please refer to lines 333 to 335 on Page 7**), all ROI definitions and spatial mappings strictly follow the established pipelines. Thus, our FC computation, use of sliding windows and multi-scale feature extraction is not discards anatomical spatial relationships, rather, it leverages the functional topology embedded in the FC matrix (**please refer to lines 164 to 171 on Page 4**) and do not introduce any unintended loss or distortion of anatomical structure.
>
>
> [1] 10.1007/978-3-030-87234-2_51
>
> [2] 10.1109/ISBI52829.2022.9761486
>
> [3] 10.1109/BIBM52615.2021.9669488
>
> [4] 10.1016/j.neunet.2024.106945
>
> [5] 10.3389/fninf.2022.769274
>
> We hope that these clarifications address your concerns. Thank you again for your thoughtful feedback.

---

### Official Review · Reviewer_cQhP · 2025-11-09

**Soundness:** 2
**Presentation:** 2
**Contribution:** 2
**Rating:** 4
**Confidence:** 3

**Summary:**

This manuscript introduces Geo-Mamba, a geometric state-space model designed to address the misspecification of conventional Euclidean sequence models when analyzing functional connectivity (FC) data from fMRI, which inherently resides on the SPD manifold. Geo-Mamba employs a dual-path selective state-space design, combining hierarchical modeling with geometry-aware dimensionality reduction, and fusing their outputs via a tailored GeoMix operator. The model was evaluated on six public fMRI datasets against several baseline methods.

**Strengths:**

- Originality: The method introduces an original combination of hierarchical pyramid multi-spatial scale FC aggregates and stacks them to form a sequence fed to a state space model, which preserves geometry.

- Evaluation: The authors evaluate their approach across several public datasets, encompassing both small and large scales, and compare it to several relevant baselines.

- Neuroscientific Interpretation: The interpretation of the results - although limited to a subpath of the entire architecture - established meaningful connections to existing neuroscientific understanding.

**Weaknesses:**

- Interpretation analysis was limited to path #2, omitting coverage of the SSM path.
- The title and introduction emphasize SSMs, yet the proposed method incorporates multiple paths and components in addition to the proposed geometric SSM, making it difficult to isolate the source of performance gains.
- The ablation analysis lacks clarity. It would be beneficial to focus on a single key metric (e.g., accuracy) and move others to the appendix. Obvious ablation candidates include removing path #1 and/or path #2 entirely. Another relevant candidate is ablating the self-attention block within path #1.
- Table 4 (appendix) indicates the use of different hyperparameters and multiple learning rates per dataset. The authors should clarify whether they tuned the learning rate of Geo-Mamba for each dataset, how they selected the parameters, and whether the same approach was applied to the baseline methods.

**Questions:**

- What is the rationale for sharing the query (Q) across all spatial scales?
- Clarify the necessity and role of the self-attention layer positioned between the two SSM blocks, as the original Mamba architecture typically omits self-attention. Include ablation studies demonstrating the impact of removing either the self-attention or the SSM blocks.
- After model fitting, analyze the learned weight alpha in the GeoMix layer to determine the relative importance of path 1 versus path 2.
- The description of the geometry preserving core SSM component (lines 210 to 220), needs significant rephrasing and expansion for clarity.
- There is an incorrect reference at line 330 (Adam et al. 2014).

---

> ### Author Response · Authors · 2025-11-21
> **Respone to Reviewer cQhP**
>
> ### Thank you for acknowledging the contributions of our work. We are thrilled and grateful for your insightful feedback, which has significantly contributed to enhancing the quality of our manuscript. In the following responses, **W** and **Q** represent Weaknesses and Questions, and **A** represents the corresponding answer.
>
> **W1**. On interpretation being limited to Path #2.
>
> **A1**:  Our interpretation analysis focuses on Path #2 because reconstructing the fused low-dimensional representation back to the original SPD space requires a stable and accurate inverse projection. Path #2 performs dimensionality reduction directly from the global SPD manifold, making its projection matrix better suited for faithful reconstruction. Path #1 involves local multi-scale aggregation, which does not support accurate SPD restoration. We have clarified this rationale in the revised manuscript (**please refer to lines 413 to 419 on Page 8** ).
>
> **W2**. On the model containing multiple components beyond the geometric SSM.
>
> **A2**:  While Geo-Mamba adopts a dual-path design, both paths operate within the proposed geometric SSM framework. The additional modules (e.g., self-attention, GeoMix) are auxiliary mechanisms for feature extraction and SSM-aware fusion rather than independent modeling pathways.
>
> **W3**.  The ablation analysis lacks clarity.
>
> **A3**:  Following the reviewer’s suggestion, we reorganized the ablations to highlight accuracy as the primary metric and moved secondary metrics to the Appendix. We also added new ablations by removing the self-attention module to better isolate each component’s contribution. **Please refer to lines 471 to 474 on Page 9** .
>
> **W4**. On dataset-specific hyperparameters and learning rates.
>
> **A4**:  For our model (Geo-Mamba), grid search was performed according to the hyperparameter ranges defined in Appendix Table 4. The Acc, Pre, and F1 scores reported reflect the optimal configuration selected from this search (**please refer to lines 355 to 357 on Page 7** ). For the baseline methods, we strictly followed the official implementations and used their recommended default hyperparameters, including learning rate. This is a standard practice in the field, and we did not re-tune baselines beyond the settings provided in their public code, in order to ensure fair and reproducible comparisons. We have clarified this procedure in the revised version (**please refer to lines 341 to 349 on Page 7** ).
>
> **Q1**.  What is the rationale for sharing the query Q across all spatial scales?
>
> **A5**:  In our model, the query Q is not shared across all spatial scales.  Specifically, the module containing Q — the geometric attention pooling(GAP) — is applied independently at each window size 𝑤.  GAP integrates the multi-scale features $Z_w$(obtained via pyramid feature extraction and dimensionality compression) into a weighted representation $P_w$, which characterizes the features corresponding to window size 𝑤.  Therefore, different window sizes have their own GAP modules and associated Q parameters. We have made it clear in the revised manuscript (**please refer to lines 188 to 189 on Page 4** ).

---

> ### Author Response · Authors · 2025-11-21
>
> **Q2**.  Clarify the role of the self-attention layer between SSM blocks.
>
> **A6**:  Thank you for your valuable comment. The self-attention layer between the two SSM blocks is essential for capturing cross-regional dependencies in the SPD matrix sequence.(**please refer to lines 290 to 295 on Page 6** ) The first SSM primarily models local spatial dynamics, while the attention module aggregates geodesic-based similarities across spatial steps, enabling the model to encode global interactions that cannot be captured by a single SSM alone. The second SSM then integrates these enriched representations, preserving both local and global low-dimensional spatial patterns for downstream classification. We have added an ablation study in the revised manuscript (**please refer to lines 471 to 474 on Page 9** ), which demonstrates that removing either the self-attention layer or one of the SSM blocks leads to a consistent performance drop, confirming their complementary roles.
>
> | Method | Metric | ADNI | OASIS | PPMI | Taowu | Neurocon | Mātai |
> |--------|--------|--------|--------|--------|--------|--------|--------|
> | remove SSM_1 | Acc | 77.60 ± 8.24 | 88.81 ± 2.04 | 70.16 ± 8.50 | 97.50 ± 7.50 | 97.50 ± 7.50 | 88.33 ± 7.64 |
> | remove SSM_1 | Pre | 74.59 ± 13.97 | 87.60 ± 2.68 | 64.07 ± 12.47 | 95.63 ± 13.13 | 98.33 ± 5.00 | 89.36 ± 8.72 |
> | remove SSM_1 | F1 | 73.61 ± 11.35 | 86.41 ± 2.07 | 63.76 ± 10.91 | 96.43 ± 10.71 | 97.33 ± 8.00 | 87.15 ± 8.66 |
> | remove self-attention | Acc | 80.40 ± 7.47 | 89.22 ± 1.94 | 67.63 ± 5.92 | 97.50 ± 7.50 | 97.50 ± 7.50 | 90.00 ± 11.06 |
> | remove self-attention | Pre | 79.79 ± 7.40 | 88.16 ± 2.73 | 63.46 ± 11.26 | 95.63 ± 13.13 | 98.33 ± 5.00 | 89.86 ± 12.19 |
> | remove self-attention | F1 | 77.02 ± 9.34 | 87.05 ± 1.55 | 59.34 ± 7.72 | 96.43 ± 10.71 | 97.33 ± 8.00 | 89.10 ± 11.80 |
> | remove dSSM_1 | Acc | 77.20 ± 9.30 | 88.89 ± 1.87 | 68.61 ± 6.79 | 90.00 ± 16.58 | 90.00 ± 16.58 | 90.00 ± 13.33 |
> | remove dSSM_1 | Pre | 72.87 ± 10.30 | 85.69 ± 4.09 | 60.46 ± 11.38 | 88.96 ± 18.64 | 83.75 ± 26.10 | 92.03 ± 10.64 |
> | remove dSSM_1 | F1 | 71.96 ± 10.18 | 86.72 ± 2.48 | 60.92 ± 8.53 | 88.76 ± 17.95 | 86.19 ± 22.56 | 88.78 ± 14.63 |
> | Geo-Mamba | Acc | **81.20 ± 6.21** | **89.39 ± 1.91** | **71.16 ± 9.12** | **97.50 ± 7.50** | **97.50 ± 7.50** | **91.67 ± 11.18** |
> | Geo-Mamba | Pre | **81.10 ± 8.12** | **88.82 ± 2.32** |**64.81 ± 10.10**| **95.63 ± 13.13** | **98.33 ± 5.00** | **88.89 ± 17.60** |
> | Geo-Mamba | F1 | **79.08 ± 7.43** | **87.10 ± 2.57** | **65.57 ± 10.17** | **96.43 ± 10.71** | **97.33 ± 8.00** | **89.48 ± 14.99** |
>
> **Q3**. relative importance of path 1 versus path 2.
>
> **A7**:  We re-analyzed the learned GeoMix weights 𝛼 to assess the relative importance of path 1 and path 2 across datasets. The fusion weights are as follows: Taowu: 0.5271, Neurocon: 0.6329, PPMI: 0.6958, Matai: 0.5191, ADNI: 0.7247, OASIS: 0.6807. These values indicate that both paths contribute meaningfully to the final representation, with path 1 generally having slightly higher influence in most datasets. We have added the analysis in the revised manuscript  (**please refer to lines 475 to 482 on Page 9** ).
>
> **Q4**  The description of the geometry preserving core SSM component (lines 210 to 220), needs significant rephrasing and expansion for clarity.
>
> **A8**:  Thank you for the valuable feedback. We have thoroughly revised and expanded the description of the core SSM components that preserve the geometric structure to improve clarity. At each step $i$, the SPD input is first mapped to the Euclidean tangent space via the logarithmic map, and the resulting vectorized representation is processed by a lightweight GRU.  The GRU output is then passed through three linear layers to produce the gating signal $\Delta_i$ and the selective operators $B_i$ and $C_i$, selective operators are projected back to the SPD manifold via Stiefel retraction to ensure orthonormality.  Unlike Mamba, which operates purely in Euclidean space, our GRU evolves in the tangent space and generates geometry-compatible operators, explicitly linking gating with SPD-preserving state updates. We believe the updated description makes this part of the manuscript significantly clearer for the reader (**please refer to lines 222 to 233 on Page 5** ).
>
> **Q5**.  There is an incorrect reference at line 330 (Adam et al. 2014).
>
> **A9**:  Sorry for the carelessness. We have updated [1] in the revised manuscript. **Please refer to lines 359 on Page 7** .
> [1] Kingma, Diederik P., and Jimmy Ba. "Adam: A method for stochastic optimization." arXiv preprint arXiv:1412.6980 (2014).

---

> > ### Comment · Reviewer_cQhP · 2025-11-23
> >
> > I appreciate the authors' detailed responses to the weaknesses and questions that I articulated.
> >
> > The new methodological details about the geometric SSM components in the revised manuscript improve clarity.
> >
> > Thank you very much for running the additional ablation experiments and summarizing them in the rebuttal. The effects seem relatively small compared to Table 1 in the manuscript (especially considering that similar/larger gaps between Geo-Mamba and baseline methods did not turn out to be statistically significant).
> >
> > It is now clear that path 1 and path 2 contribute substantially to the model predictions, which reduces the impact of the proposed model interpretation approach.
> >
> > I have not further questions/feedback. While I appreciate the authors effort in the rebuttal, I am afraid that the provided additional information did not change my overall judgement.

---

> > > ### Author Response · Authors · 2025-11-24
> > >
> > > Dear Reviewer cQhP,
> > >
> > > We really appreciate the time you spent responding to our rebuttal, especially taking the extra time over the weekend.
> > >
> > > Since you mentioned that you no longer have further questions or concerns, **we would like to kindly ask whether you might consider revisiting your overall score.** Throughout the rebuttal, we have made every effort to directly address each of your concerns, both through additional methodological clarification and new ablation experiments, and we have also incorporated your comments and suggestions into the revised manuscript. And we hope these revisions meaningfully improve the clarity and rigor of the paper.
> > >
> > > We sincerely appreciate your consideration ☺️.
> > >
> > > Best,
> > >
> > > Authors

---

### Meta-Review · Area_Chair_J9ZN · 2026-01-06

**Summary:**

A general feeling is that the method combines several existing idea into a FC-fMRI pipeline, but there is an unclear motivation or empirical demonstration of the components of the architecture, where the ablations do not seem to reveal the importance of components.
The apparent complexity and lack of clarity in methodological led the reviewers to question guiding principles behind the manuscript.

**Reviewer Concerns:**

There was much discussion about the factors important for the pipeline, and the overall need of the complexity of the pipeline. This discussion did not reach a strong conclusion, other than a need for clarifying writing of the manuscript.

**Reviewer Scores:**

The reviewers might have raised a tiny bit their score, but overall the discussions did not leave reviewers convinced.

---

### Decision · Program_Chairs · 2026-01-26

Reject